# Calendar effects on surface air temperature and precipitation based on model-ensemble equilibrium and transient simulations from PMIP4 and PACMEDY

Xiaoxu Shi[1], Martin Werner[1], Carolin Krug[1,2], Chris M. Brierley[3], Anni Zhao[3], Endurance Igbinosa[1,2], Pascale Braconnot[4], Esther Brady[5], Jian Cao[6], Roberta D'Agostino[7], Johann Jungclaus[7], Xingxing Liu[8], Bette Otto-Bliesner[5], Dmitry Sidorenko[1], Robert Tomas[5], Evgeny M. Volodin[9], Hu Yang[1], Qiong Zhang[10], Weipeng Zheng[11], and Gerrit Lohmann[1,2]

[1]Alfred Wegener Institute, Helmholtz Center for Polar and Marine Research, Bremerhaven, Germany
[2]Bremen University, Bremen, Germany
[3]Department of Geography, University College London, London, UK
[4]Laboratoire des Sciences du Climat et de l'Environnement-IPSL, Unité Mixte CEA-CNRS-UVSQ, Université Paris-Saclay, Orme des Merisiers, Gif-sur-Yvette, France
[5]Climate and Global Dynamics Laboratory, National Center for Atmospheric Research (NCAR), Boulder, CO 80305, USA
[6]School of Atmospheric Sciences, Nanjing University of Information Science & Technology, Nanjing, 210044, China
[7]Max Planck Institute for Meteorology, Hamburg, Germany
[8]State Key Laboratory of Loess and Quaternary Geology, Institute of Earth Environment, Chinese Academy of Sciences, Xi'an, 710061, China
[9]Marchuk Institute of Numerical Mathematics, Russian Academy of Sciences, ul. Gubkina 8, Moscow, 119333, Russia
[10]Department of Physical Geography and Bolin Centre for Climate Research, Stockholm University, 10691, Stockholm, Sweden
[11]LASG, Institute of Atmospheric Physics, Chinese Academy of Sciences, Beijing, 100029, China

**Correspondence:** Xiaoxu Shi (xshi@awi.de)

**Abstract.** Numerical modelling enables a comprehensive understanding not only of the Earth's system today, but also of the past. To date, a significant amount of time and effort has been devoted to paleoclimate modeling and analysis, which involves the latest and most advanced Paleoclimate Modelling Intercomparison Project phase 4 (PMIP4). The definition of seasonality, which is influenced by slow variations in the Earth's orbital parameters, plays a key role in determining the calculated seasonal cycle of the climate. In contrast to the classical calendar used today, where the lengths of the months and seasons are fixed, the angular calendar calculates the lengths of the months and seasons according to a fixed number of degrees along the Earth's orbit. When comparing simulation results for different time intervals, it is essential to account for the angular calendar to ensure that the data for comparison is from the same position along the Earth's orbit. Most models use the classical "fixed-length" calendar, which can lead to strong distortions of the monthly and seasonal values, especially for the climate of the past. Here, by analyzing daily outputs from multiple PMIP4 model simulations, we examine calendar effects on surface air temperature and precipitation under mid-Holocene, last interglacial, and pre-industrial climate conditions. We conclude that: (a) The largest cooling bias occurs in boreal autumn when the classical calendar is applied for the mid-Holocene and last interglacial, due to the fact that the vernal equinox is fixed at 21th March. (b) The sign of the temperature anomalies between the Last interglacial and pre-industrial in boreal autumn can be reversed after the switch from classical to angular calendar, particularly over the Northern

Hemisphere continents. (c) Precipitation over West Africa is overestimated in boreal summer and underestimated in boreal autumn when the "fixed-length" seasonal cycle is applied. (d) Finally, monthly-adjusted values for surface air temperature and precipitation are very similar to the daily-adjusted values, therefore correcting the calendar based on the monthly model results can largely reduce the artificial bias. In addition, we examine the calendar effects in 3 transient simulations for 6-0 ka by AWI-ESM, MPI-ESM, and IPSL-CM. We find significant discrepancies between adjusted and unadjusted temperature values over continents for both hemispheres in boreal autumn. While for other seasons the deviations are relatively small. A drying bias can be found in the summer monsoon precipitation in Africa (in the "fixed-length" calendar), whereby the magnitude of bias becomes smaller over time. Overall, our study underlines the importance of the application of calendar transformation in the analysis of climate simulations. Neglecting the calendar effects could lead to a profound artificial distortion of the calculated seasonal cycle of surface air temperature and precipitation. One important fact to be noted here is that the discrepancy in seasonality under different calendars is an analysis bias and highly depends on the choice of the reference position/date (usually the vernal equinox) on the Earth's ellipse around the sun. Different modelling groups may apply different reference dates, so ensuring a consistent reference date and seasonal definition is key when we compare results across multiple models. In most models the vernal equinox is set to 21th March, when the Sun is exactly above the Equator which leads to equal length of day and night on our Earth.

## 1 Introduction

Long-term fluctuations exist in the earth's orbital elements that affect the amount of solar radiation received by our planet (Berger, 1978). There are three parameters controlling the motion of the Earth: eccentricity, obliquity and precession. The shape of the Earth's orbit varies over time from nearly circular with a small eccentricity of 0.0034 to slightly elliptical (large eccentricity of 0.058) with major periodicities of about 400,000 and 100,000 years (Berger, 1978; Berger and Loutre, 1991). When the eccentricity is large, there is also a big difference between the perihelion distance and the aphelion distance, while at a small eccentricity when the orbit is more circular this difference is less pronounced. Earth's orbital eccentricity is 0.016764, 0.018682, and 0.039378 in 1850 CE (pre-industrial), 6 ka B.P. (mid-Holocene) and 127 ka B.P. (Last interglacial) respectively. The seasons are caused by the tilt of the Earth's axis, which is called obliquity. Boreal summer occurs when the Earth's North Pole is tilted toward the sun, and vice versa when boreal winter prevails. Earth's axial obliquity oscillates between 22.1 and 24.5 degrees with a major period of 41,000 years. A high obliquity results in stronger seasonal cycles than a low obliquity does. At the same time, the wobble of Earth's rotational axis (precession) modifies the direction of the Earth's tilt and determines which hemisphere is tilted towards the sun at perihelion. The major periodicities of climatic precession are around 19,000 and 23,000 years (Berger, 1978). Precession determines the beginning of each season relative to Earth's orbit and therefore has a

major impact on the seasonal pattern of solar radiation. Understanding the role of the three elements of Earth's orbit can help us better examine and interpret past climates from seasonal to millennial time scales.

Numerical modeling of the past climate, which is very different from today, can in many aspects improve our understanding of the underlying mechanisms of the Earth's system and help us better predict the future climate. The Paleoclimate Model Intercomparison Project (PMIP) brings together a number of modelling groups, providing the ability to synchronize results from different models (Kageyama et al., 2018, 2021a).

Two interglacial episodes, i.e., the mid-Holocene (MH, a period roughly from 7 to 5 ka B.P.) and the Last interglacial (LIG, roughly equivalent to 130-115 ka B.P.), are particularly the focus of PMIP (Otto-Bliesner et al., 2017), as they are the two most recent warm periods in geological history. So far, there are a variety of previous studies aiming to examine the simulated climate of mid-Holocene and Last interglacial. Due to the Earth's orbital parameter anomalies with respect to the present, the MH and LIG receive more insolation in boreal summer and less in boreal winter over the Northern Hemisphere, leading to larger seasonal contrast in the two time periods, which holds true for both hemispheres in most model simulations (Kukla et al., 2002; Shi and Lohmann, 2016; Shi et al., 2020; Zhang et al., 2021; Kageyama et al., 2021b; Herold et al., 2012). Such effect is much more profound in the LIG than in the MH (Lunt et al., 2013; Pfeiffer and Lohmann, 2016). However, in earlier simulations using CCSM3 and LOVECLIM, Nikolova et al. (2013) found smaller seasonality across Lazarev Sea (in CCSM3) and South Atlantic Ocean (in LOVECLIM) during the Last interglacial as compared to PI. Climate models identified a northward shift of the Intertropical Convergence Zone (ITCZ) during the two periods, accompanied by a northward displacement of the Northern Hemisphere monsoon domains (Jiang et al., 2015; Braconnot et al., 2007; Nikolova et al., 2013; Fischer and Jungclaus, 2010; Herold et al., 2012). The precession of the MH and LIG, which determines the length of each season, was also different from today. Following the orbital definition of seasons, this results in a calendar (hereafter referred to as angular calendar) that is different from today's calendar (hereafter referred to as classical calendar). It has been pointed out in Joussaume and Braconnot (1997) that significant biases occur when we apply today's classical calendar to the MH and LIG seasonal cycles. Therefore, it is important to consider the orbital configuration when defining seasonal cycles for past climate. However, the calendar effect has been investigated in only a few paleoclimate studies. Differences of seasonal ensemble anomalies (LIG minus PI) based on the angular and the classical calendars have been shown by Scussolini et al. (2019) for both precipitation and surface air temperature. Their results indicated pronounced artificial bias for the classical calendar definition: The Northern Hemisphere warming (LIG minus PI) in boreal summer is largely underestimated. Moreover, the Northern Hemisphere monsoon precipitation during the LIG is overestimated in boreal summer but underestimated in boreal autumn. These results are in line with the findings of Joussaume and Braconnot (1997). A recent study by Bartlein and Shafer (2019) examined the "pure" responses of temperature and precipitation to calendar conversion; this was accomplished by applying angular calendars of 6, 97, 116 and 127 ka in a modern climate state. Our present study differs from Bartlein and Shafer (2019) in the following aspects: 1. We use daily data instead of monthly data, so a more accurate result is guaranteed. 2. We perform calendar correction for pre-industrial as well, as today's Gregorian calendar is not an angular one. It should be noted that in most previous studies today's calendar has been left unchanged (Joussaume and Braconnot, 1997; Bartlein and Shafer, 2019). 3. In Bartlein and Shafer (2019), the "pure" calendar effects have been examined by applying the angular calendar of 6 ka, 97

ka, 116 ka, and 127 ka onto modern observations. In the present study, we perform a calendar adjustment based on the actual past time intervals of the different model experiments. In detail, we apply an angular calendar of 0 ka, 6 ka, and 127 ka for the pre-industrial, mid-Holocene, and Last interglacial simulation respectively.

In the present study, we use the PMIP4 dataset to investigate the calendar effect on the simulated surface air temperatures and precipitation under MH and LIG boundary conditions. The structure of the paper is as follows: In Section 2, we describe the method for defining an angular calendar based on the Earth's orbital parameters and provide detailed information on the data we used. In Section 3 we first briefly describe the main features of simulated MH and LIG surface air temperatures and precipitation, then we illustrate the effects of the angular season definition on the simulated patterns. We discuss and conclude in Section 4 and 5, respectively.

## 2 Methodology

### 2.1 Calendar correction

In order to appropriately compare the seasonal climate between different time periods resonating with the respective orbital configuration, the seasonality should be calculated according to the position of the Earth along its orbit. First, we define the true anomaly $\theta$ as the angle between the axis of the perihelion and the actual position of the earth. Note that the term "anomaly", standing for "angle", is used in astronomy to describe planetary positions. We then define a month (season) as a 30 (90 degree) increment of the true anomaly, integrated from a fixed starting point. The vernal equinox (VE) is set on March 21 at noon. In the following, we compute the length of a month (season) by calculating how much time the Earth needs to move from the respective starting point to the endpoint. For this purpose, we derive the relation between the true anomaly of any given time and the time elapsed since the Earth passes perihelion.

We define the mean anomaly $M$ as the angle between the perihelion and Earth's position based on the assumption that the orbit describes a perfect circle with the sun at the center by:

$$M = \frac{2\pi}{T} \cdot t_p \tag{1}$$

Here, $t_p$ denotes the time elapsed since Earth passes the perihelion and $T$ is the Earth's revolution period (i.e., 1 year or 365 days), namely the time it takes the Earth to make one complete revolution around the sun. Taking into account the orbit's eccentricity $\epsilon$, we define the eccentric anomaly $E$ via:

$$E - \epsilon \cdot \sin(E) = M \tag{2}$$

Equation (2) is called Kepler's equation and is based on Kepler's 1st and 2nd laws (Fig. S1). The first law simply states that the orbit of a planet is an ellipse with the Sun at one of the two focus points, and the Kepler's 2nd law states that a line segment connecting the sun and a planet sweeps out equal areas during equal intervals of time. Equation (2) can be solved with the

application of Newton's method. For more detailed information we refer to Danby and Burkardt (1983). E can be found using the following expression (Eq. 3.13b of Curtis (2014)):

$$E = 2 \cdot \arctan(\sqrt{\frac{1 - \epsilon}{1 + \epsilon}} \cdot \tan(\frac{\theta}{2})) \tag{3}$$

The above equations implicitly relates $t_p$ to $\theta$ by:

$$t_p(\theta) = \frac{MT}{2\pi} = \frac{(E - \epsilon \cdot \sin(E))T}{2\pi} \tag{4}$$

Note that E is defined in Equation (3).

The relation between the true anomaly $\theta$ and the time elapsed since Earth passes perihelion $t_p$ allows to define seasons with respect to Earth's position on the orbit rather than relying on a fixed number of days. Based on the "fixed-angular" approach, there are two ways to define the seasons: 1) The orbit is distinguished into four segments: A true anomaly of $\theta = 0°$) corresponds to March 21st and therefore marks the first day of boreal spring. The length of the boreal summer is gained by
calculating $t_p$ ($\theta = 90°$). Similarly, the terms $t_p$ ($\theta = 180°$) and $t_p$ ($\theta = 270°$) mark the beginning of boreal fall and winter, respectively. 2) The other method is based on the "meteorological" definition, in which the boreal spring is defined as March-April-May, as typically done in paleoclimate modelling, although the VE is set to March 21st. The second approach is adopted in our study, and in this case, we firstly compute the starting and end time for each month, then average over the respective months in order to compare the angular seasonal means with the classical seasonal means. Months can be defined as 30°
increments of the true anomaly. Just one additional step has to be executed before calculating angular months: As no months starts at the VE, the starting day has to be shifted from March 21st to April 1st. Since the time between today's March 21st and April 1st may not be true for past calendars, we defined April 1st by the angle. Therefore, we first calculate the angle between today's March 21st, noon (the VE) and the point of time occurring 10.5 days later, denoting April 1st. Finally, starting from the angle corresponding to April 1st, we are able to calculate the starting time of the next month by 30° increments of
the true anomaly. Here we apply the so-called "largest remainder method": the number of days defined by the 30° of true longitude usually consists of an integer part plus a fractional remainder. Each month is firstly allocated a number of days equal to its respective integer part (for example, if January has 31.76 days, 31 days are allocated). This generally leave some days unallocated. The months are then ranked according to their fractional remainders, then an additional day is allocated to each of the months with the largest remainders until all days have been allocated.

The calendar correction method can only be suitably applied on daily data. If only monthly data is available, an alternative option is to reconstruct the daily time series in a way that original monthly mean averages are preserved, then to perform calendar conversion based on the reconstructed daily time series. The mean preserving algorithm is presented in Rymes and Myers (2001).

**Table 1.** List of PMIP4 model data used in the present study.

| Name | Institution | Reference | Notes |
|---|---|---|---|
| AWI-ESM-1-1-LR | AWI | Sidorenko et al. (2015); Rackow et al. (2018) | Dynamic vegetation |
| AWI-ESM-2-1-LR | AWI | Sidorenko et al. (2019) | Dynamic vegetation |
| CESM2 | NCAR | Gettelman et al. (2019) | Potential Natural Land Cover |
| EC-Earth3-LR | Stockholm University | | Prescribed vegetation and aerosols |
| FGOALS-f3-L | IAP-CAS | He et al. (2019) | daily precipitation for PI is missing |
| FGOALS-g3 | IAP-CAS | Li et al. (2020) | - |
| INM-CM4-8 | INM RAS | Volodin et al. (2018) | Prescribed vegetation, simulated aerosols |
| IPSL-CM6A-LR | IPSL | Lurton et al. (2020) | Prescribed vegetation, interactive phenology, prescribed PI aerosols |
| NESM3 | NUIST | Cao et al. (2018) | - |

**Table 2.** PMIP4 boundary conditions for pre-industrial, mid-Holocene and Last interglacial.

| Experiment | $CO_2$ (ppm) | $CH_4$ (ppb) | $N_2O$ (ppb) | Eccentricity | Obliquity | perihelion - 180° |
|---|---|---|---|---|---|---|
| PI | 284.3 | 808.2 | 273 | 0.016764 | 23.459° | 100.33° |
| MH | 264.4 | 597 | 262 | 0.018682 | 24.105° | 0.87° |
| LIG | 275 | 685 | 255 | 0.039378 | 24.040° | 275.41° |

## 2.2 Data

We collect the PMIP4 models which provide daily outputs of surface air temperature and precipitation for equilibrium simulations of pre-industrial, mid-Holocene and Last interglacial. There are 9 models that meet the requirement, and we list the detailed information of those models in Table 1.

According to Otto-Bliesner et al. (2017), the $CO_2$ concentration applied in the PMIP4 protocol for mid-Holocene is derived from ice-core measurements from Dome C (Monnin et al., 2001, 2004). $CH_4$ has been derived from multiple Antarctic ice

cores including EPICA Dome C (Flückiger et al., 2002), EPICA Dronning Maud Land (Barbante et al., 2006) and Talos Dome (Buiron et al., 2011).The $N_2O$ data around 6 ka are compiled from EPICA Dome C (Flückiger et al., 2002; Spahni et al., 2005) and Greenland ice cores. The concentrations of $CO_2$ during the LIG are derived from Antarctic ice cores (Bereiter et al., 2015; Schneider et al., 2013), $CH_4$ has been derived from EPICA Dome C and EPICA Dronning Maud Land (Loulergue et al., 2008; Schilt et al., 2010b), and $N_2O$ from EPICA Dome C and Talos Dome (Schilt et al., 2010b, a). Table 2 provides a summary of

PMIP4 boundary conditions for pre-industrial, mid-Holocene and Last interglacial.

Besides equilibrium simulations, we also use the monthly surface air temperature and precipitation from 3 transient simulations for the past 6,000 years, based on the Earth system models AWI-ESM, MPI-ESM, and IPSL-CM. Using AWI-ESM,

we firstly conducted a 1,000-year mid-Holocene simulation with dynamic vegetation which was used as initial conditions for the transient experiment. We then conducted the 6-0 ka transient experiment, by applying the boundary conditions of the past 6,000 years with the last year representing 1950 CE. Orbital parameters are calculated according to Berger (1977), and the greenhouse gases are taken from ice-core records and from recent measurements of firn air and atmospheric samples (Köhler et al., 2017). The transient simulation performed by MPI-ESM spans the period from 6000 BP until 1850 CE, and was initialized from a previous mid-Holocene equilibrium simulation. The model is forced by prescribed orbitally-induced variations in the insolation following Berger (1977). $CO_2$, $CH_4$ and $N_2O$ forcings stem from ice-core reconstructions (Brovkin et al., 2019). The model accounts for dynamic vegetation changes in the land-surface model JSBACH. A more detailed description of the boundary conditions and the forcing of the transient simulation are given in Bader et al. (2020). The IPSL-CM transient simulation was initialized from a 1,000-year mid-Holocene spin-up run. The Earth's orbital parameters are derived from Berger (1977), the concentrations of the trace gases ($CO_2$, $CH_4$ and $N_2O$) are set based on reconstruction from ice core data (Joos and Spahni, 2008), and the vegetation was calculated interactively within the model. More detailed information about the IPSL-CM transient simulation can be found in Braconnot et al. (2019). Therefore, in the transient simulations, the orbital forcings used at 6 ka and 0 ka are the same as the PMIP4 equilibrium simulations. However, there are differences between the greenhouse gas concentrations applied in the transient and PMIP4 equilibrium simulations, as the values have been taken from different reconstructions.

## 3 Results

### 3.1 Climate responses to the MH and LIG boundary conditions under classical calendar

Owing to the altered orbital parameters, the MH receives more (less) incoming solar radiation over the Northern Hemisphere during boreal summer (winter) than present (Fig. S2a). As a consequence, the MH Northern Hemisphere experiences a cooling (up to -2 K) and warming (up to 2.5 K) in DJF and JJA respectively (Fig. S3a-b). For the annual average, our model ensemble reveals a general cooling (Fig. S3c) over Northern Hemisphere, which seems to be inconsistent with the increased annual mean insolation forcing. This phenomenon can be explained by the decreased concentration of greenhouse gases in the MH as compared to present-day condition, which leads to an effective radiative forcing of about -0.3 W/m2, as estimated by Otto-Bliesner et al. (2017).

Regarding the Southern Hemisphere we observe a general cooling in DJF (Fig. S3a), dominated by the decreased insolation in January and February (Fig. S2a). The warming across the Southern Ocean is due to a delayed effect of the increased solar energy in SON. Due to the large heat capacity of water, the ocean responses much more slowly to changes in incoming insolation than the land. Therefore, changes in solar radiation and surface air temperature over the oceans are out of phase. During the MH, the Southern Hemisphere receives more radiation flux in SON in relative to present-day, leading to a warming of the Southern Ocean in DJF. Moreover, the models present a robust cooling over most regions of Southern Hemisphere in JJA, which is mainly led by the reduction in greenhouse gases as the difference in the incomming solar radiation between the MH and PI is negligible.

The changes in surface air temperature in the LIG with respect to the PI, as shown in Fig. S3d-f, are much more pronounced than that between the MH and the PI. The most intriguing feature is an enhancement in seasonality during the LIG, with a DJF cooling being up to -5 K (over North Africa and South Asia), as well as a JJA warming (more then 5 K) over North America and Eurasia. This is mostly contributed by the corresponding anomalies in solar insolation (Fig. S2c). In addition, the model-ensemble produces a cooling over Sahel region as a response to the intensification in monsoonal rainfall. For the Southern Hemisphere, the subtropical continents also experience a DJF cooling and JJA warming (more then 2 K) as responses to the altered incoming solar radiation. Such feature is robust across the models.

The summer monsoon precipitation is shown to be enhanced over the Northern Hemisphere monsoon domains, in both MH and LIG as compared to modern condition (Fig. S3g-l), driven by the changes in seasonal insolation and the northward displacement of the Inter Tropical Convergence Zone (ITCZ). The monsoon domain at northern Africa, as well as South Asia, expands significantly in the LIG in relative to PI, associated with a stronger land-sea thermal contrast, and an intensification of moisture transport during monsoon seasons. Our results in terms of the responses of the surface air temperature and precipitation to the MH and LIG boundary conditions are in good agreement with the results from the full PMIP4 ensemble as described in Brierley et al. (2020), Otto-Bliesner et al. (2021), and Scussolini et al. (2019), as well as the studies of earlier PMIP ensemble simulations (Lunt et al., 2013).

### 3.2 Shifts in months/seasons between classical and angular calendars

The calculated duration of the angular months and seasons is shown in Table 3. For PI, the shifts in the beginning of most months between the classical and angular calendar are generally in the range of -1 to 2 days, with the exception of October with a 3-day shift. So for today the two approaches are similar. Since the orbital velocity of the Earth is greater at perihelion than at aphelion, the seasons at aphelion are longer than at perihelion, for example for the present-day we have fewer days in boreal winter and more days in boreal summer, which is reflected both in today's classical calendar (DJF: 90 days; JJA: 92 days) and in the angular calendar (DJF: 89 days; JJA: 93 days). The shifts of months for MH are in the range of -2 to 3 days, and the largest shift occurs mainly in the boreal winter. In the MH, boreal winter and spring are longer in the angular calendar than in the classical calendar, while boreal summer and autumn are shorter. Due to the large difference in precession in the LIG compared to today, there are significant shifts in the beginning of the months between classical and angular calendars, especially in boreal autumn (about -10 days). During the LIG, boreal winter has 98 days when the angular calendar is used, which is much longer than boreal summer (85 days).

### 3.3 Calendar effects in equilibrium simulations

### 3.3.1 Surface air temperature

Now we turn to examine the calendar effects on the seasonal cycle of surface air temperature. Fig. 1 depicts the differences in seasonal surface air temperature between angular and classical means. Positive/negative values indicate warming/cooling in angular mean temperatures as compared to classical mean temperatures. We observe spatially-variable changes of surface air

**Table 3.** Starting and end date of angular month in PI, MH and LIG, referencing to today's classical calendar in a no-leap year, calculated based on the approach described in Section 2.1.

| Month/Season | PI | MH | LIG |
|---|---|---|---|
| Jan. | 02Jan.-30Jan. | 29Dec.-28Jan. | 26Dec.-27Jan. |
| Feb. | 31Jan.-01Mar. | 29Jan.-28Feb. | 28Jan.-28Feb. |
| Mar. | 02Mar.-31Mar. | 01Mar.-31Mar. | 01Mar.-31Mar. |
| Apr. | 01Apr.-01May. | 01Apr.-02May. | 01Apr.-30Apr. |
| May. | 02May.-01Jun. | 03May.-02Jun. | 01May.-29May. |
| Jun. | 02Jun.-02Jul. | 03Jun.-03Jul. | 30May.-26Jun. |
| Jul. | 03Jul.-02Aug. | 04Jul.-02Aug. | 27Jun.-24Jul. |
| Aug. | 03Aug.-02Sep. | 03Aug.-01Sep. | 25Jul.-22Aug. |
| Sep. | 03Sep.-03Oct. | 02Sep.-30Sep. | 23Aug.-20Sep. |
| Oct. | 04Oct.-02Nov. | 01Oct.-29Oct. | 21Sep.-21Oct. |
| Nov. | 03Nov.-02Dec. | 30Oct.-28Nov. | 22Oct.-22Nov. |
| Dec. | 03Dec.-01Jan. | 29Nov.-28Dec. | 23Nov.-25Dec. |
| Boreal winter | 03Dec.-01Mar. (89) | 29Nov.-28Feb. (92) | 23Nov.-28Feb. (98) |
| Boreal spring | 02Mar.-01Jun. (92) | 01Mar.-02Jun. (94) | 01Mar.-29May. (90) |
| Boreal summer | 02Jun.-02Sep. (93) | 03Jun.-01Sep. (91) | 30May.-22Aug. (85) |
| Boreal autumn | 03Sep.-02Dec. (91) | 02Sep.-28Nov. (88) | 23Aug.-22Nov. (92) |

temperature in adjusted values as compared to unadjusted values. For the LIG, the most pronounced pattern is a warming over the Northern Hemisphere up to 5 K in boreal autumn (SON), as well as a cooling over the Southern Hemisphere especially the Antarctic continent (up to -3 K). This is explicable by the fact that the angular SON receives more/less insolation over Northern/Southern Hemisphere than the classical SON does (Fig. 2a, Fig. S2), in agreement with the earlier onset of those months. As the VE is fixed at March 21st, the calendar effect is expected to be relatively minor for boreal spring (MAM). Indeed, we find only slight increase (within 0.3 K) in the Northern Hemisphere surface air temperature in classical means as compared to angular means, and for the Southern Hemisphere the calendar-adjusted minus unadjusted values are in the range of -0.1 to 0 K, dominated by the pattern in May (Fig. S4). We are aware of that there is no difference between adjusted and unadjusted values in March and April, as no shift occurs in the beginning and duration of these two months (Table 3). In boreal winter (DJF), the most prominent calendar effects on LIG surface air temperature can be seen over Northern Hemisphere, with a warming up to 1.5 K, as well as the oceans of Southern Hemisphere which experiences a cooling up to -0.4 K. Such pattern is dominated by the temperature anomalies in December (Fig. S4). The warming signal over Antarctica (0-0.5 K) in DJF, is mainly determined by increased insolation during January and February. The conversion of the calendar produces a cooling (within -1 K) over Northern Hemisphere ocean and Southern Hemisphere continents (except Antarctic) in boreal summer (JJA), while for other regions, especially the Northern Hemisphere continents, we obtain positive anomalies in surface air temperature.

Compared to the LIG, the response of surface air temperature to calendar effect in the MH are less pronounced (Fig. 1). It reveals a dipole pattern in all seasons, with warming over the Northern Hemisphere and cooling over the Southern Hemisphere. One exception is the Antarctica warming in boreal winter, led by the increased insolation in January and February over Antarctica (Fig. 2b). Fig. S5 shows the adjusted minus non-adjusted temperatures for each month. No difference is found for March as for mid-Holocene the beginning and end of March in the angular calendar are the same as in the modern classical calendar (Table 3). From April to June, the delay in the angular calendar leads to a positive insolation difference and therefore a warming over the Northern Hemisphere, while the opposite case is for the Southern Hemisphere. Similar patterns are observed for October to November, but is due to an advance in those months (the peak insolation happens in June). In general, we notice that the temperature anomalies on continents are in phase with the insolation changes, while the calendar effect on surface air temperature over the ocean is delayed due to the large heat capacity of sea water.

For PI, the classical "fixed-length" calendar used at present is similar to today's angular calendar from January to June (Table 3), this leads to relatively minor changes in surface air temperature in boreal winter, spring and summer (Fig. 1i-k, Fig. S6) in angular mean values as compared to classical mean values. In boreal autumn, a dipole pattern of insolation anomaly is obvious (Fig. 2c): less (more) insolation is received at the top of the atmosphere over Northern (Southern) Hemisphere in adjusted SON than that in non-adjusted SON, consistent with the delay of boreal autumn in angular calendar as compared to classical calendar. Such a pattern favors cooling (up to -0.4 K) over the Northern Hemisphere and warming over the Southern Hemisphere during SON.

Knowing the pure calendar effect on the surface air temperature for respective time period, now we turn to investigate to what degree the temperature anomalies between paleo and pre-industrial can be affected by calendar conversion. As shown by Fig. 3, in boreal winter, spring, and summer, we observe similar patterns for both definitions of seasonal means. The insolation changes induced by changes in orbital parameters produce an enhanced seasonality in LIG as compared to PI, with colder boreal winter and warmer boreal summer, especially over Northern Hemisphere continents. However, with classical calendar applied, the DJF cooling over Northern Hemisphere is overestimated by up to 1 K. Whilst an underestimation in the MAM cooling happen over Northern Hemisphere, with a magnitude up to 1 K. For JJA, the bias in temperature anomaly, as calculated from classical means, is not uniform and has a clear land-sea contrast. Classical calendar tends to underestimate the JJA warming over Northern Hemisphere lands (by 1 K) and Southern Hemisphere oceans (0.2 K); while the warming over North Atlantic, North Pacific, as well as Southern Hemisphere continents are overestimated in classical calendar. The most prominent calendar effect can be seen in SON, as the temperature anomaly over Northern Hemisphere continents in SON flips its sign after switching from classical means to angular means, with the magnitude of the bias being as large as -5 K for classical means. Such artificial bias could be interpreted as climatic signals without the application of the adjusted calendar.

For the temperature anomalies between MH and PI as shown in Fig. 4, the most significant bias introduced by the use of the classical calendar occurs in SON over Northern Hemisphere continents (more than 1 K), which appears to be colder in MH as compared to PI for classical means, and warmer for angular means. Moreover, the warming over Antarctica in MH relative to PI is overestimated in the classical calendar. From DJF through MAM, both calendars show a general colder-than-present climate in MH, and the use of the present classical calendar causes a cooling bias (within -0.5 K) for the Northern Hemisphere

and Antarctic, as well as a warming bias (within 0.3 K) for the Southern Hemisphere oceans. In boreal summer, the key characteristic shared in both angular and classical means is a warming over the Arctic Ocean, North Atlantic and Eurasia, led by increased JJA insolation in MH as compared to PI. Such warming is more pronounced in angular calendar than in classical calendar.

Analysis on individual models reveals a robust calendar effect on SON surface air temperature for both continents and oceans, which overwhelms the differences between models (Fig. 5). We also observe that the calendar effect on temperature anomalies is more pronounced at higher latitudes than at lower latitudes.

### 3.3.2 Precipitation

In LIG, the largest calendar effects on precipitation can be observed for SON over the tropical rain-belt (Fig. 6 shows the anomalies and Fig. S7 shows the percentage changes), with positive anomalies (within 30 mm/month) to the north and negative anomalies (up to -30 mm/month) to the south of the Inter Tropical Convergence Zone (ITCZ). In North Africa, changes in precipitation due to calendar transition account for up to 80% of the classical mean (Fig. S7d). In DJF, we observe a tripole pattern, with negative anomalies over North (-1 mm/month, -10%) and South Africa (-4 mm/month, -5%) and positive anomalies over equatorial Africa (5 mm/month, 8%). For JJA the adjusted-minus-unadjusted precipitation anomalies present a dryness (up to -15 mm/month, -15%) and wetness (less than 10 mm/month, 16%) over the northern and southern edge of the ITCZ, respectively, opposite to the patterns for SON and DJF. The calendar effect appears to be small during boreal spring, as the vernal equinox is fixed at 21 March in both calendars. In contrast to the calendar-induced significant changes in large-scale patterns of LIG precipitation, the effect of calendar on MH precipitation is much less pronounced, showing positive (negative) anomalies up to 5 (-5) mm/month over the north (south) branch of the tropical rain-belt for all seasons. This is associated with the di-pole pattern of temperature differences between angular and classical means (warming over Northern Hemisphere and cooling over Southern Hemisphere). For PI, a northward displacement of ITCZ is obvious during SON for angular mean as compared to classical mean precipitation. While, for other seasons, no pronounced changes in precipitation can be observed.

The anomalies in precipitation (LIG-PI), as well as the impact of calendar conversion on the precipitation anomalies are shown in Fig. 7. The general patterns of precipitation anomalies (LIG-PI) are very similar for both angular and classical means, revealing a northward shift of the ITCZ especially from JJA through SON, evidenced in the wetter condition to the north of ITCZ and the drier condition to the south. Such pattern is overestimated in JJA and underestimated in SON when the present classical calendar is applied. For both calendars, MH also presents a similar distribution in precipitation anomalies as for LIG, with a much smaller magnitude (Fig. 8). Moreover, the application of the classical calendar leads to an underestimation of the increased summer monsoon rainfall in MH as compared to PI over the Northern Hemisphere monsoon domains, i.e., West Africa, North America, and South Asia.

Fig. 9 depicts the calendar impact on the SON precipitation anomaly over the main monsoon domains of the Northern Hemisphere (i.e. North America, North Africa and South Asia). We notice a very large model-model discrepancy for all regions examined in both the MH and the LIG, with the exception of North Africa in the MH. Our results indicate that that during the MH, the precipitation in South Asia is more responsive to a calendar adjustment compared to North Africa and

North America. However, for the LIG, no robust conclusion could be drawn about the calendar effects in the different regions due to the large discrepancies between the models.

Overall, it is crucial to perform calendar conversion before examining the surface temperature and precipitation differences between LIG/MH and PI, as non-ignorable artificial bias can be introduced to the seasonal cycle of temperature and precipitation with the application of present classical calendar, which could be misinterpreted as climatic feedbacks.

### 3.3.3 Calendar conversion based on monthly data

Daily output takes up much more space than monthly output, so most modelling groups only provide monthly frequency variables. Here, we utilize a calendar transformation method that requires only the raw (i.e., classical "fixed-length" calendar) monthly mean values (Rymes and Myers, 2001). In the study of Rymes and Myers (2001) an approach has been introduced for smoothly interpolating coarsely-resolved data onto a finer resolution, while preserving the deterministic mean. Based on the approach, daily data can be reconstructed using the monthly mean values: The daily data is initialised with the monthly average of the respective month. Then, for each day of the year, its value is recursively recalculated as the average of its own value and the values of the two adjacent days. After 365 iterations, this results in a nicely smooth annual cycle with the original monthly means being preserved. Using this approach, we perform calendar corrections based on the monthly outputs of the same 9 modelling groups. We then check the deviation of this month-length adjusted values from the day-length adjusted values. Here the month-length and day-length adjusted values represent the adjusted values after calendar correction based on the original daily and monthly data respectively. From Fig. S8 and Fig. S9 we can conclude that the conversion of calendar based on monthly mean values can improve the seasonal cycle to a large degree. For MH and PI, we observe only slight bias, with the temperature deviation being less than $\pm$ 0.05 K and precipitation deviation less than $\pm$ 1 mm/month, indicating that the calendar transformation based on monthly data can serve as an alternative for seasonal adjustment of MH and PI. We are aware of a slight artificial bias in month-length adjusted surface air temperature for LIG over the high-latitude continents in JJA, which is underestimated by 0.07 K. During boreal autumn, the land is generally cooler and the tropics and Southern Ocean are generally warmer compared to the day-length adjusted values.

As stated above, we find spatial heterogeneity in the response of surface air temperature to calendar conversion across the globe, which is manifested in the opposite signals between two hemispheres and the contrast between land and ocean. Our model ensemble shows that the calendar effect is more pronounced over continents than over seawater areas. Here we calculate the seasonal cycle of surface air temperature for: (1) the original daily average, (2) the original monthly average, (3) daily length-adjusted mean values, and (4) month-length adjusted mean values, over different continents, shown in Fig. 10. We find the day-length and month-length adjusted values are very similar, evidenced in the overlapping orange and purple solid lines in Fig. 10. This suggests that the monthly calendar correction approach can serve as a good alternative when only monthly frequency model outputs are available for surface air temperature. For North America and Eurasia, we observe a slight positive anomaly in the PI between adjusted and unadjusted surface temperatures from January to July (less than 0.2 K), while negative anomalies are found from August to December, with the maximum change occurring in October (-0.7 K). For the Antarctic, the greatest calendar effect occurs in October and November, with the mean adjusted-minus-unadjusted value being 0.8 K.

This agrees with the spatial maps shown in Fig. S6. For the MH, the calendar effect over North America and Eurasia appears to be greatest in May-June (0.5 K) and October-December (0.6 K). Over the Antarctic continent, apart from the warming in January-February (0.5 K) and the cooling in November (-0.7 K), no significant response of the mean surface air temperature to the calendar conversion was found. In terms of the LIG, the mean adjusted-minus-unadjusted surface air temperature in October reached up to 3 K in both North America and Eurasia. The maximum temperature change in Antarctica also occurs in October, with a magnitude of -3 K. In addition, we calculated the seasonal cycle of precipitation values for the following monsoon domains: North America (5-30°N, 120-40°W), African monsoon region (5-23.3°N, 15W-30°E), and South Asia (5-23.3°N, 70°W-120°E). As shown in Fig. 11, again we see very similar day-length and month-length adjusted values. Therefore, performing calendar correction based on monthly precipitation can help reduce the artificial distortion of monsoon rains to a large extent.

### 3.4    Calendar effects in transient simulations

Calendar effects should be considered also in the analysis of transient simulations (Bartlein and Shafer, 2019). Here with the utility of 3 mid-Holocene-to-present transient runs based on AWI-ESM, MPI-ESM and IPSL-CM respectively, we examine the degree of influence of calendar definition on surface air temperature and precipitation. All the three experiments provide outputs in monthly frequency, therefore we perform calendar transformation based on monthly surface air temperature and precipitation using the approach described by Rymes and Myers (2001).

The time series plotted in Fig. 12 are for adjusted and unadjusted mean surface air temperature over the Northern Hemisphere continents (i.e., Greenland, North America, Eurasia, and northern Africa) for all seasons. Based on all the 3 models, the largest deviation between angular and classical mean temperature values happens in boreal autumn between 6 and 4.4 ka, with the temperature being underestimated under the present "fixed-length" calendar. Another distinct difference between month-length adjusted and unadjusted values occurs in boreal autumn between 4.4 and 0 ka. During this time interval, the surface air temperature over Northern Hemisphere continents can be overestimated when using the classical calendar. This phenomenon, again, highlights the importance of calendar correction in the analysis of both mid-Holocene and pre-industrial climates, especially in boreal autumn. Without the calculation of angular seasonality, the warming in the mid-Holocene relative to pre-industrial in SON can be largely underestimated. In DJF, no obvious deviation is found between the angular and classical means, evidenced in the overlapped black and red lines in the top panels of Fig. 12. During boreal spring, all the 3 models reveal a slight cooling bias in the original temperature values throughout the whole integrated time period, which is relatively more manifested in the mid-Holocene and pre-industrial than in 3-1 ka. In JJA, besides the slight cooling bias in the original mean surface air temperature for 6-3 ka as revealed by all the 3 models, we observe a model-dependency of the calendar effects for the time interval of 3-0 ka, during which the Northern Hemisphere classical mean temperature in JJA is slightly underestimated by AWI-ESM and MPI-ESM, but for IPSL-CM the adjust and unadjusted values are identical. Such discrepancy between models is related to the spatially varying temperature changes over the Northern Hemisphere continents caused by the calendar effect (Fig. 1k). The calendar effect on Northern Hemisphere temperature over oceans, as shown in Fig. 13, is very similar to that over lands. However, the deviation between adjusted and unadjusted SON temperature is much less pronounced. This is also

consistent with the results from the equilibrium simulations. Moreover, in JJA, all models show positive anomalies of the angular-minus-classial mean temperature over ocean, which the magnitudes being smaller from 6 to 0 ka..

For the Southern Hemisphere lands, including South America, Australia, Southern Africa, and Antarctic, as shown in Fig. 14, the calendar effects are less pronounced as compared to Northern Hemisphere. Similar to the Northern Hemisphere, no distinct temperature deviation is seen for DJF. Besides, all the three models agree on the cooling bias in classical-mean temperatures in SON from 4 to 0 ka, as well as a slight warming deviation during MAM (6-0 ka). For JJA, no noticeable change in temperature could be found in IPSL, while the other two models (AWI-ESM and MPI-ESM) reveal a positive anomaly between the adjusted and unadjusted means. The oceans appear to have a more pronounced response to calendar adjustment in boreal autumn (Fig. 15). For other seasons, no obvious deviation of temperature is seen for the Southern Hemisphere oceans.

Fig. S10 illustrated the calendar effects on the Africa monsoon precipitation. The time series in Fig. S10 are derived by averaging month-length adjusted and unadjusted JJA precipitation over the land points within 5-23.3°N, 15W-30°E. All the 3 transient simulations show a slight artificial drying bias in the Africa monsoon precipitation with the application of the "fixed-length" calendar in 6 ka. It is also shown that, such calendar effect gradually becomes weaker from mid-Holocene to present.

# 4   Discussion

Two important elements should be taken into consideration when comparing paleoclimate simulations of different time intervals: the reference date (usually the VE), and the angle of the orbit of the Earth around the Sun, which defines the phasing of the insolation curve. Artificial bias emerges when precessional effects are ignored, and such bias can be amplified by eccentricity changes (Joussaume and Braconnot, 1997). To avoid such bias, one shall define the seasonal cycle based on astronomical positions along the elliptical orbits. The sensitivity of simulated paleoclimate conditions to the "classical" and "angular" calendars had been investigated in a former study based on one single coarse resolution model (Joussaume and Braconnot, 1997), in which the authors state that the differences between the two calendar means cannot be neglected. Here by examining 7 of the most advanced climate models in PMIP4, we again confirm the necessity of calendar definition in paleoclimate modelling research.

Daily data is needed for calendar adjustment, however, due to the large volume of daily outputs, they are not preserved by most modelling groups. A mean preserving algorithm has been introduced (Rymes and Myers, 2001), with which the daily time series can be reconstructed. By performing calendar correction on the reconstructed daily time series, we find that the seasonal pattern in temperature and precipitation can be largely ameliorated, even though there is still room for improvement.

Various methods for adjusting monthly data towards an angular calendar have been suggested. Rymes and Myers (2001) developed a mean-preserving running-mean algorithm to reconstruct the annual cycle. In Pollard and Reusch (2002), the reconstruction of an annual cycle was based on a spline method, which fits each monthly segment by a parabola, requiring the same monthly means as the originals and continuity of value and slope at the month boundaries. Bartlein and Shafer (2019), used a mean-preserving harmonic interpolation method described in Epstein (1991) and performed the same function as the

parabolic-spline interpolation method as in Pollard and Reusch (2002). To sum up, the basic procedure is similar in all the

approaches, as they are all based on "mean-preserving" algorithm. In Bartlein and Shafer (2019), a comparison was made between the linear and mean-preserving interpolation methods. They found that the difference between the original monthly means and the monthly means of the linearly interpolated daily values is not negligible for both surface air temperature and precipitation while the difference between an original monthly mean value and one calculated using the mean-preserving interpolation method is negligible.

In previous studies, the angular calendar was defined using the true anomaly of the Earth corresponding to the present-day seasons, in other words, each month begins and ends at the same celestial longitude as present-day for any period (Joussaume and Braconnot, 1997; Bartlein and Shafer, 2019; Timm et al., 2008; Chen et al., 2011; Pollard and Reusch, 2002). The work of Chen et al. (2011) and Timm et al. (2008) applied a 360-day year which is, originally, divided into 12 months with 30 days. The VE is set to day 81 in a calendar year. Pollard and Reusch (2002), Joussaume and Braconnot (1997) and Bartlein and Shafer (2019), on the other hand, performed the calendar adjustment based on today's classical calendar with 365 days in a non-leap year. In their studies, an assumption was made that the seasonality defined by the classical calendar is in phase with the insolation and solar geometry for modern-day. In our study, by calculating the onset of present-day months/seasons using the approach described in Section 2.1, we find that the classical "fixed-length" calendar is very similar to the angular calendar for today, but they are not completely the same. This is evidenced in the small shift of months between the two calendars as seen in Table 3. In particular the angular October is delayed by 3 days compared to the classical October, resulting in negative anomalies in the adjusted-minus-unadjusted solar insolation. Though different methods are used in our work from the mentioned previous studies, our results are identical: for the LIG, the adjusted-minus-unadjusted surface air temperature over the Northern Hemisphere is up to 5 K during SON (Joussaume and Braconnot, 1997; Bartlein and Shafer, 2019; Chen et al., 2011) or September (Pollard and Reusch, 2002); and the Northern Hemisphere monsoon precipitation in SON is underestimated by the use of the classical calendar (Bartlein and Shafer, 2019; Chen et al., 2011). Similar biases are found for the early-Holocene (Timm et al., 2008) and mid-Holocene (Joussaume and Braconnot, 1997; Bartlein and Shafer, 2019) but less pronounced . These results are consistent with the findings in our study, however, comparing results of our 3 transient simulations with that from the TraCE-21ka transient simulation, as it was investigated in Bartlein and Shafer (2019), distinct differences emerge for the boreal autumn surface air temperature near present-day. In Bartlein and Shafer (2019), the artificial bias in MH-minus-PI temperature and precipitation totally stems from the bias in MH when the classical calendar is applied (as for PI both calendars are identical). In contrast, our study reveals that such bias is mainly dominated by the deviation between angular and classical calendars for present-day. It should be noted that these discrepancies are not due to the different models used in our studies, but rather to the different approaches adopted for calendar adjustment.

An interesting phenomenon shared by our model-ensemble transient simulations and TraCE-21ka (Bartlein and Shafer, 2019) is that, around 6k all seasons show an increased surface air temperature over Northern Hemisphere continents in angular means compared to angular means (Fig. 12). The annual mean temperature should, however, be the same regardless of the seasonality definition used. This is due to the different lengths of seasons between the two approaches. Therefore, our results support the strategy as described in Zhao et al. (2021): when averaging modeled variables across multiple months/seasons, it

is desirable to perform calendar correction and to take into consideration the lengths of each month/season in order to avoid

extra artificial bias introduced by the calculation, or directly use the daily output if available.

Proxy-based reconstructions provide us another ability to examine the temperature evolution of the past and can help assess the model's performance in simulating the past climates. Since paleoclimate data often records the seasonal signal (e.g. local summer temperature), an appropriate choice of calendar is therefore important for temperature comparisons between model results and proxy data. For the mid-Holocene, Bartlein et al. (2011) is an often-cited study that compiled pollen-based conti-

nental temperature reconstructions. The question arises whether the consideration of calendar effects could lead to an improved model-data agreement. Here we show in Fig. S11 the simulated classical mean temperature anomalies (MH minus PI) versus continental reconstructions. The expected increased seasonality occurs only over Northwest Europe as indicated by the proxy records. The opposite sign is shown over northern America, with winter warming and summer cooling, and is therefore not consistent with the ensemble model result. Bartlein et al. (2011) attributes such a model-data mismatch to changes in local

atmospheric circulation that tend to overwhelm the insolation effect. The calendar impacts, as illustrated in Fig. 4, result in warming of less than 0.2 K over the Northern Hemisphere in both DJF and JJA, implying that model-data consistency is improved for Northwest Europe in boreal summer, and Northern America in winter, while for most other regions using the adjusted calendar results in a poorer match between model and proxy temperatures. These results reveal that for the mid-Holocene the calendar adjustment does not guarantee a better model-data agreement, and the underlying reason might be that,

in addition to the solar insolation, the proxy could be strongly influenced by the local environment, such as flow of humid air and increased cloud cover (Harrison et al., 2003) or warm-air advection (Bonfils et al., 2004).

Since there are very few high-resolution reconstructed temperature records for the LIG, we use here the compilation from Turney and Jones (2010) for the annual mean temperature anomalies between LIG and PI, and compare them with modeled classical mean values for boreal summer (Fig. S12). We keep in mind that the Northern Hemisphere summer mean LIG

temperatures are usually higher than the annual mean values documented by the proxy records. At high latitudes of Northern Hemisphere continents (e.g. Greenland, Russia and Alaska), as well as over subpolar oceans (e.g. the Nordic Sea and the Labrador Sea), we find that the models underestimate the recorded LIG warming. Part of the bias can be corrected by calendar adjustment which leads to a warming of up to 1 K over Northern Hemisphere continents in JJA (Fig. 3k).

Not all types of archives are sensitive to calendar definition, for instance bioclimatic indicators might be less dependent on

the artificial definition of seasons, a typical example here is the the growing degree-days (GDD). In addition, we examined the influence of the calendar effect on the simulated vegetation. For this we analyzed the simulated leaf area index. As revealed by Fig. S13, even during boreal autumn, the deviation in leaf area index between classical and angular calendars is below 0.06% for PI and MH, and below 0.2% for LIG. Therefore, the calendar effect plays no significant role for this vegetation-related variable.

Finally, we should bear in mind that the forcing or boundary conditions of the paleoclimate simulations may still indirectly include a reference to today's calendar (e.g., prescribed monthly data of ozone, vegetation or aerosols). This is particularly important for paleoclimate simulations with stand-alone atmosphere or ocean models, as they are often forced by fields based

on a classical calendar, and this may introduce further a bias in the simulated seasonality even if the calendar effect has been considered.

## 5 Conclusions

In the present paper, we use March 21st as the reference VE date, and perform calendar correction for 3 climatic periods: the pre-industrial, the mid-Holocene, and the Last interglacial. The results indicate that the precessional effects are the strongest in the Last interglacial, with the strongest effect for boreal autumn. In boreal autumn, the classical mean Northern Hemisphere temperature in Last interglacial has a severe cooling bias, which largely impacts the anomaly between Last interglacial and
pre-industrial. Similar case is also found for mid-Holocene, just with a less pronounced magnitude. It should be pointed out that, even though today's season lengths are in phase with the orbital definition of seasons, today's calendar is not an angular calendar. To be consistent, today's calendar also needs to be corrected, and this leads to non-ignorable changes in boreal autumn.

Another indication from the present paper is that the calendar definition can greatly affect the calculated Africa monsoon
rainfall in the LIG, which starts from late June and ends in October (Zhang and Cook, 2014; Sultan and Janicot, 2003). We find that using a classical calendar leads to overestimation (underestimation) of Africa monsoon rainfall in boreal summer (autumn). Therefore, consideration of the calendar conversion is very essential for investigating the Africa monsoon precipitation during the LIG.

Finally, our results support the method of calendar adjustment based on monthly model output, which is shown to be able
to largely reduce the artificial bias in surface air temperature and precipitation, and can therefore serve as an alternative of the daily data-based calendar conversion approach.

*Code availability.* The Python source code and related manual are available from https://gitlab.awi.de/xshi/calendar (last access 02.02.2022).

*Data availability.* The daily model outputs from the PMIP4 equilibrium simulations used in the present study can be downloaded from the Earth System Grid Federation (ESGF): https://esgf-data.dkrz.de/projects/esgf-dkrz/. The data from the transient simulations can be accessed
by contacting XS (AWI-ESM), RD (MPI-ESM), and PB (IPSL-CM).

*Author contributions.* XS, GL and EI developed the original idea for this study. XS conducted the AWI-ESM experiments, produces all figures and wrote the initial draft under the supervision of MW and GL. CK wrote the scripts for calendar correction, and drafted the methodology section. EI performed calendar correction and first analysis under the supervision of XS and GL. CK, CB, AZ, PB and XL gave very constructive comments and suggestions. DS provided technical help of AWI-ESM. RD and JJ performed MPI-ESM transient simulation.

PB performed IPSL-CM transient simulation. EB, JC, BO, RT, EV, HY, QZ and WZ performed key PMIP4 equilibrium experiments and contributed their respective model outputs.

*Competing interests.*  The authors have declared that no competing interests exist.

*Acknowledgements.*  We would to express our appreciation for the constructive comments from Pepijn Bakker and another two anonymous reviewers. The present study is supported by Alfred Wegener Institute, Helmholtz center for Polar and Marine Research; the PAlaeo-Constraints
on Monsoon Evolution and Dynamics (PACMEDY) Belmont Forum project, grand number 01LP1607A; German Federal Ministry of Education and Science (BMBF) PalMod II WP 3.3 (grant no. 01LP1924B); The AWI-ESM simulations were conducted on Deutsche Klimarechenzentrum (DKRZ) and AWI supercomputer (Ollie). XL is funded by the open fund of State Key Laboratory of Loess and Quaternary Geology, Institute of Earth Environment, CAS (grant no. SKLLQG1920) and the National Science Foundation for Young Scientists of China (grant no. 41807425); RD is funded by the Deutsche Forschungsgemeinschaft (DFG, German Research Foundation) under Germany's Excellence
Strategy – EXC 2037 Climate, Climatic Change, and Society (CLICCS) - Cluster of Excellence Hamburg, A4 African and Asian Monsoon Margins – Project Number: 390683824; QZ is funded by the Swedish Research Council (Vetenskapsrdet, grant no. 2013-06476 and 2017-04232). The EC-Earth simulations were performed on HPC resources provided by the Swedish National Infrastructure for Computing (SNIC) at the National Supercomputer Centre (NSC).

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

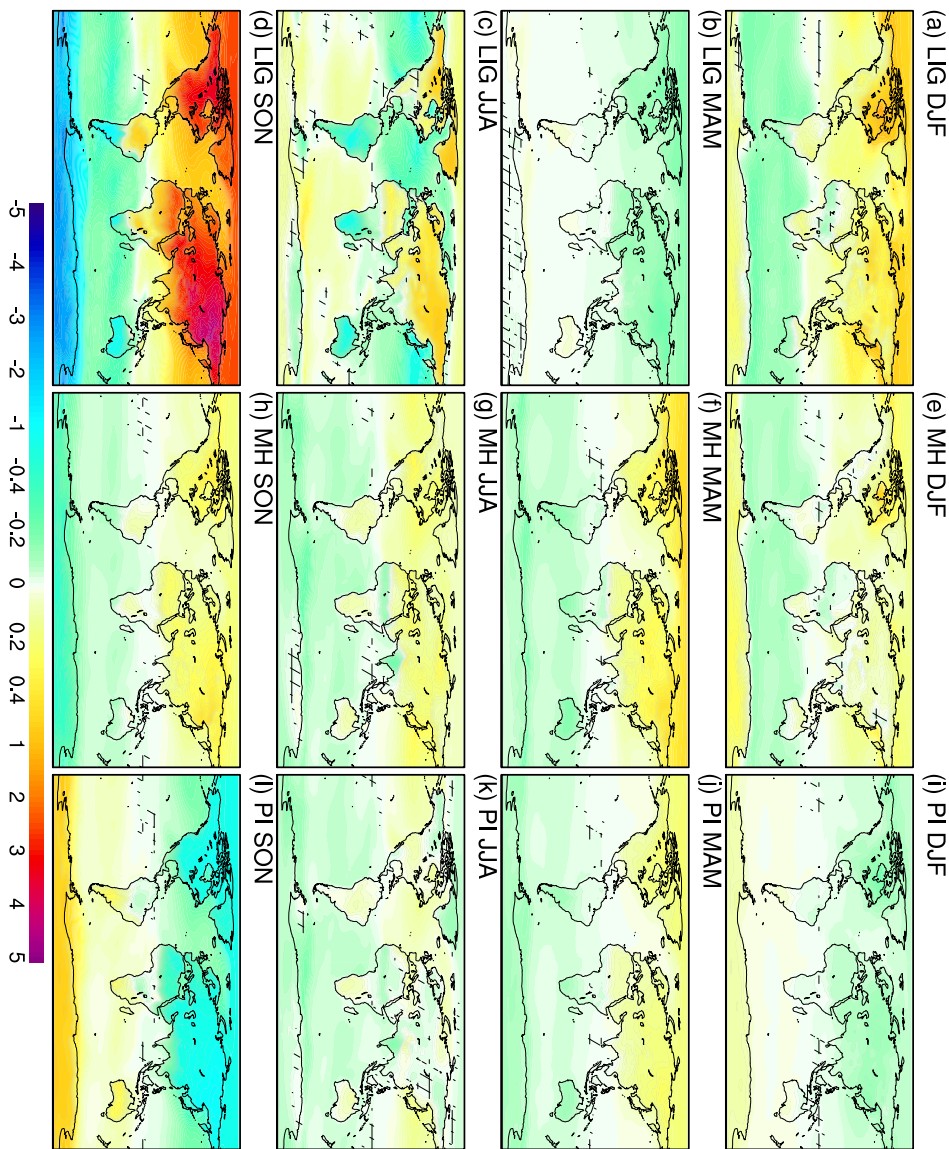

**Figure 1.** Ensemble anomalies of surface air temperature between angular means and classical means. The unmarked area indicates that at least 7 models show the same sign. Units: K.

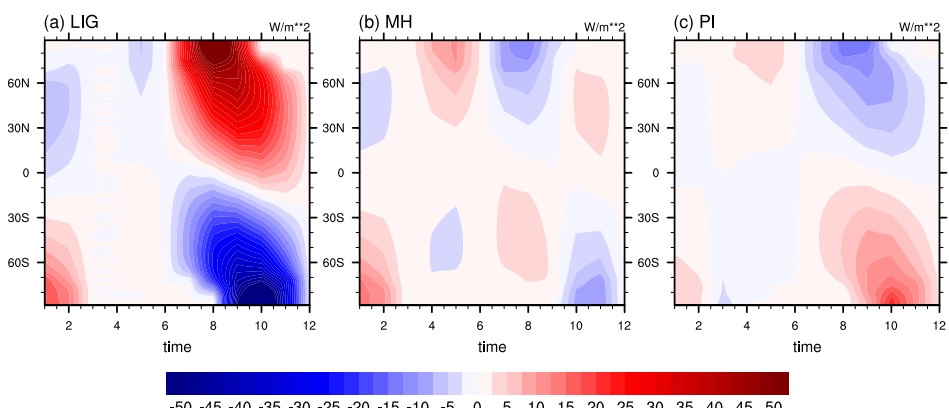

**Figure 2.** Insolation anomalies between angular and classical calendar for (a) LIG, (b) MH, and (c) PI.

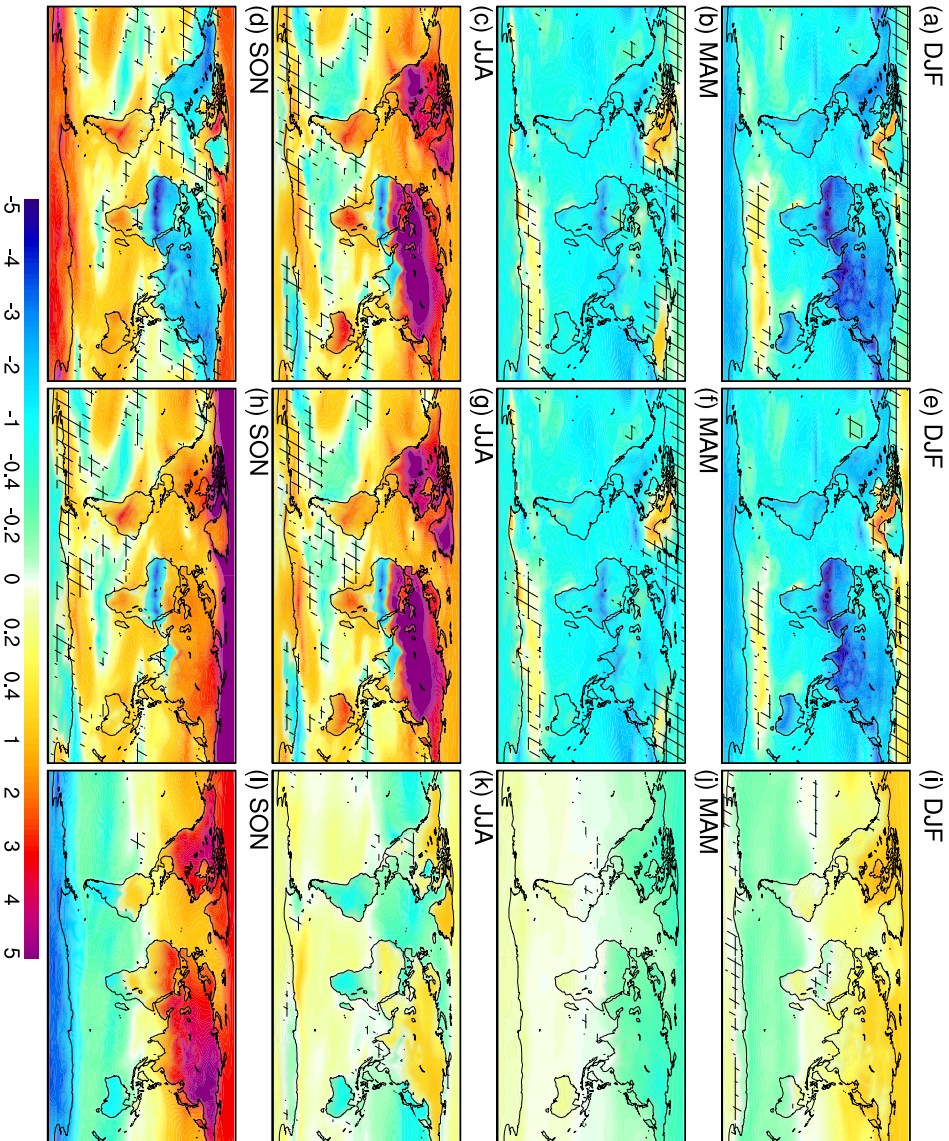

**Figure 3.** Ensemble surface air temperature for (a-d) LIG minus PI classical means, (e-h) LIG minus PI angular means, and (i-l) anomalies between LIG minus PI angular means and LIG minus PI classical means. The unmarked area indicates that at least 7 models show the same sign. Units: K.

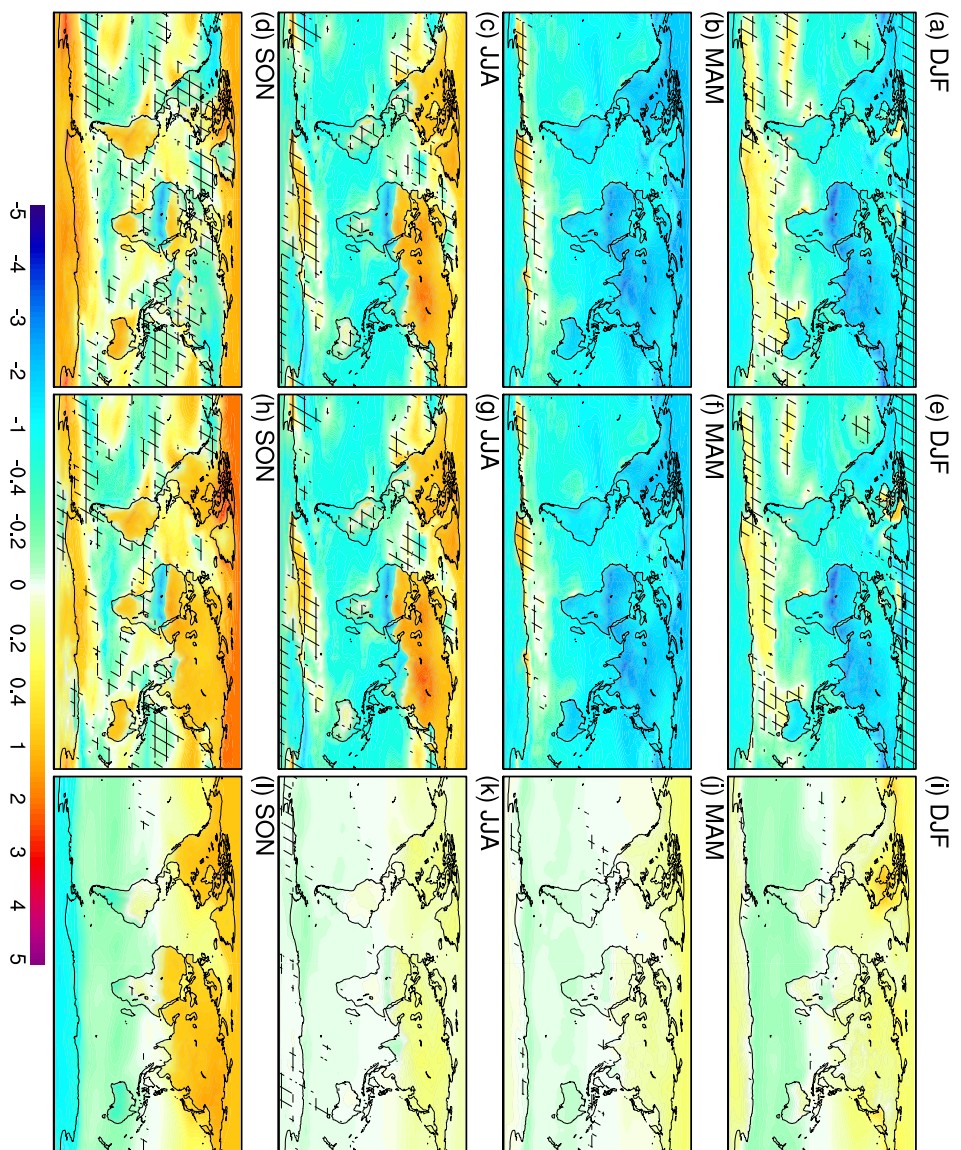

**Figure 4.** As Fig. 3, but for the MH.

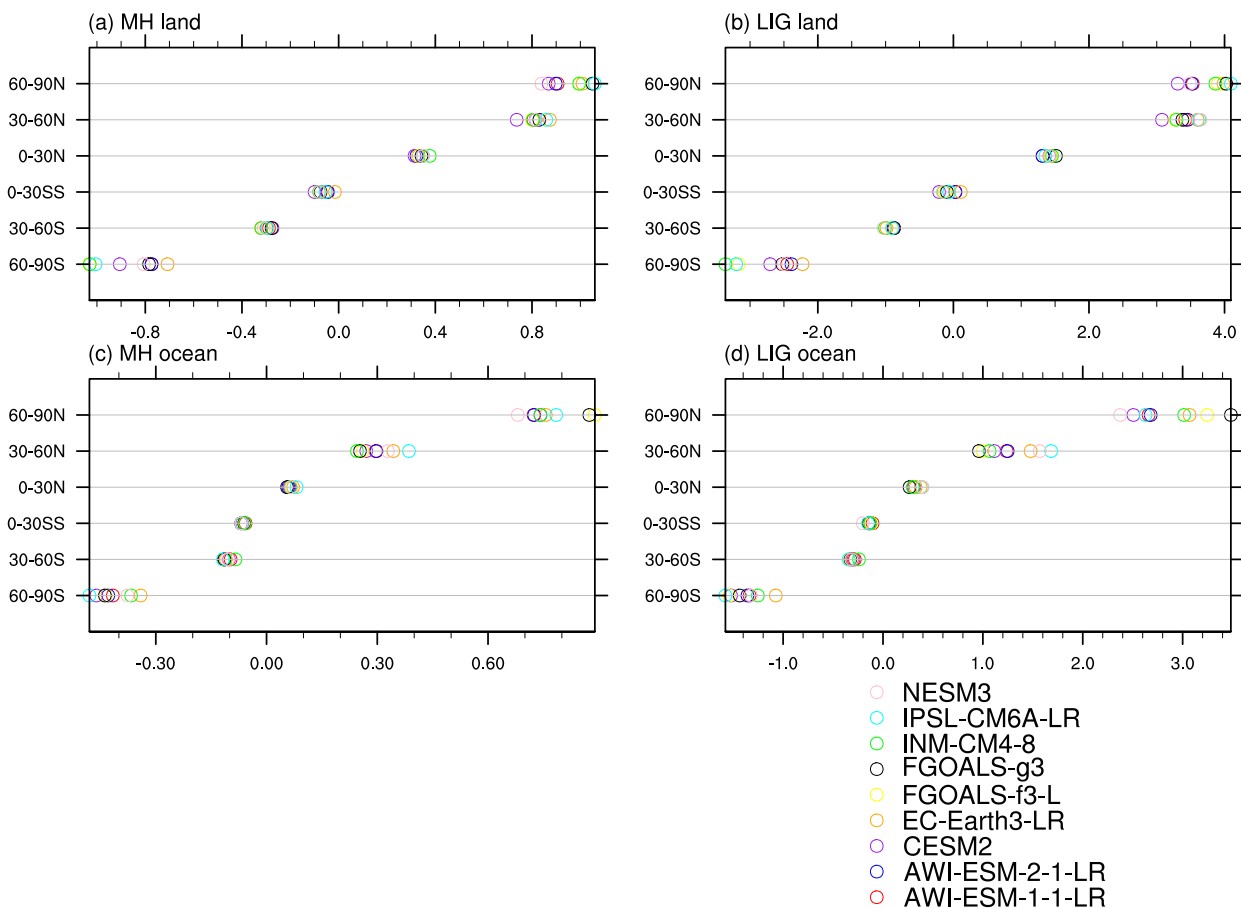

**Figure 5.** (a,b) Deviation of MH-PI SON surface air temperature between angular and classical means for (a) continents and (b) oceans at different latitude-bands, simulated by individual models. (c,d) As in (a,b), but for LIG-PI surface air temperature. Units: K.

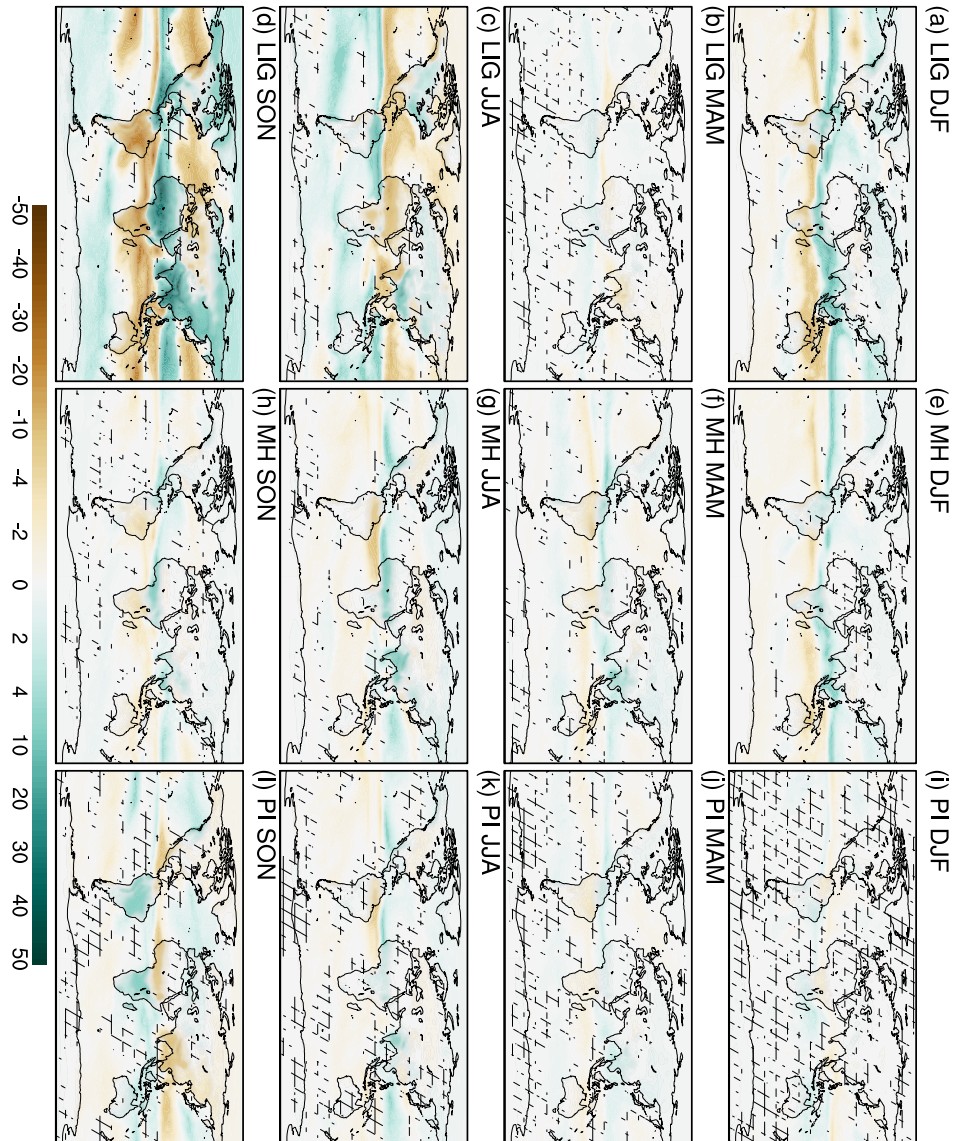

**Figure 6.** Ensemble anomalies of precipitation between angular and classical means for (a-d) LIG, (e-h) MH, and (i-l) PI. The unmarked area indicates that at least 7 models show the same sign. Units: mm/month.

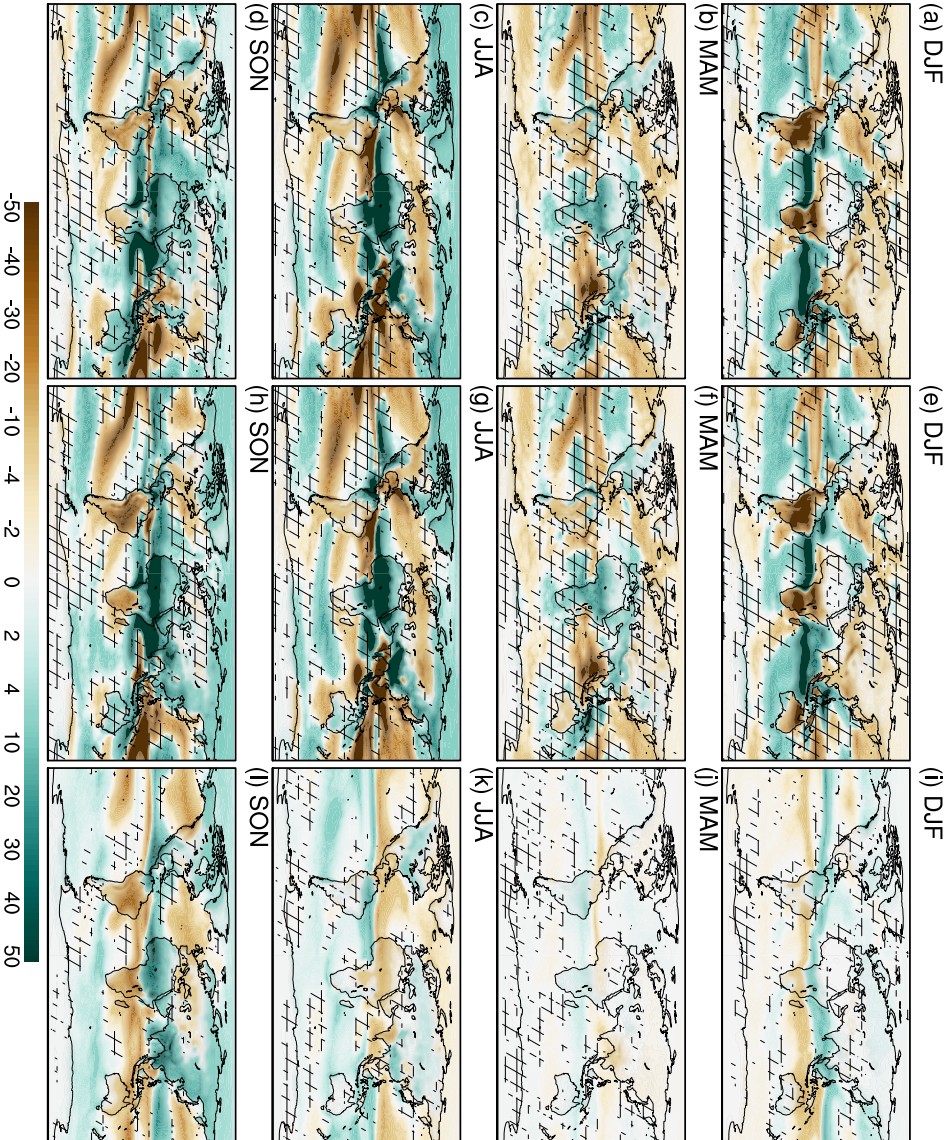

**Figure 7.** Ensemble precipitation for (a-d) LIG minus PI classical means, (e-h) LIG minus PI angular means, and (i-l) anomalies between LIG minus PI classical means and LIG minus PI angular means. The unmarked area indicates that at least 7 models show the same sign. Units: mm/month.

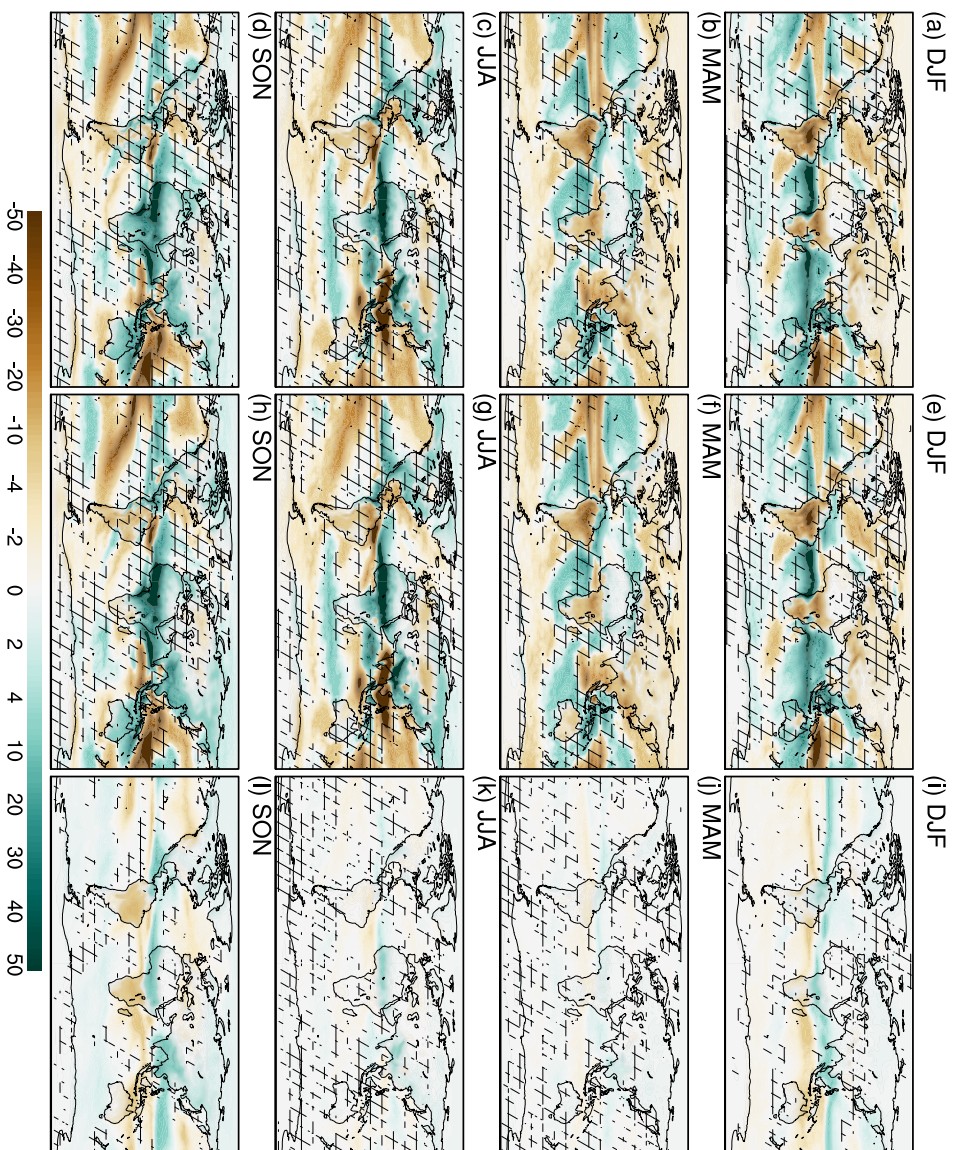

**Figure 8.** As Fig. 7, but for the MH.

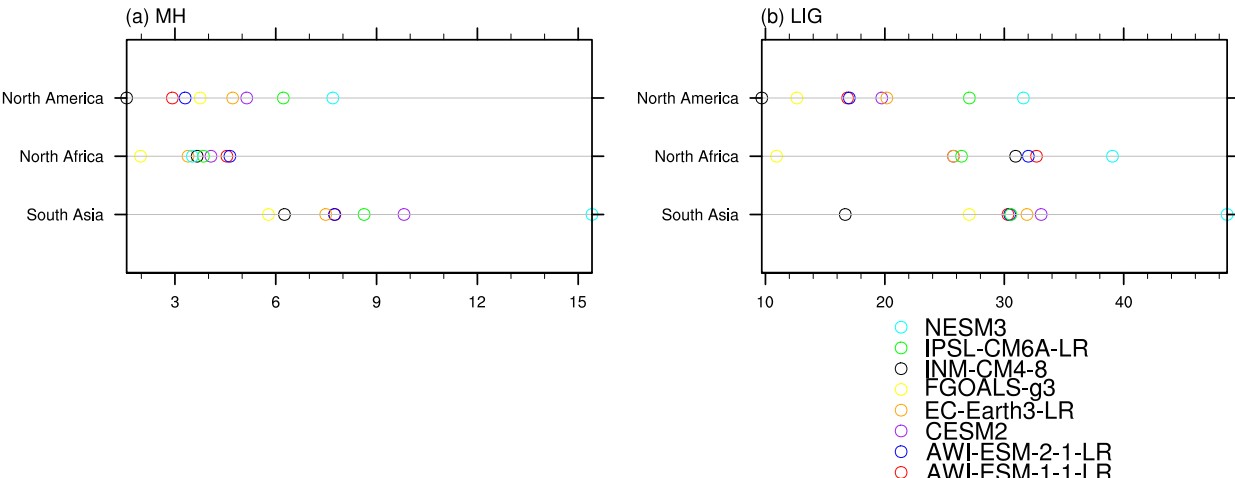

**Figure 9.** (a) Deviation of SON MH-PI precipitation between angular and classical means for North America, North Africa and South Asia, simulated by individual models. (b) As in (a), but for LIG-PI precipitation. Units: mm/month.

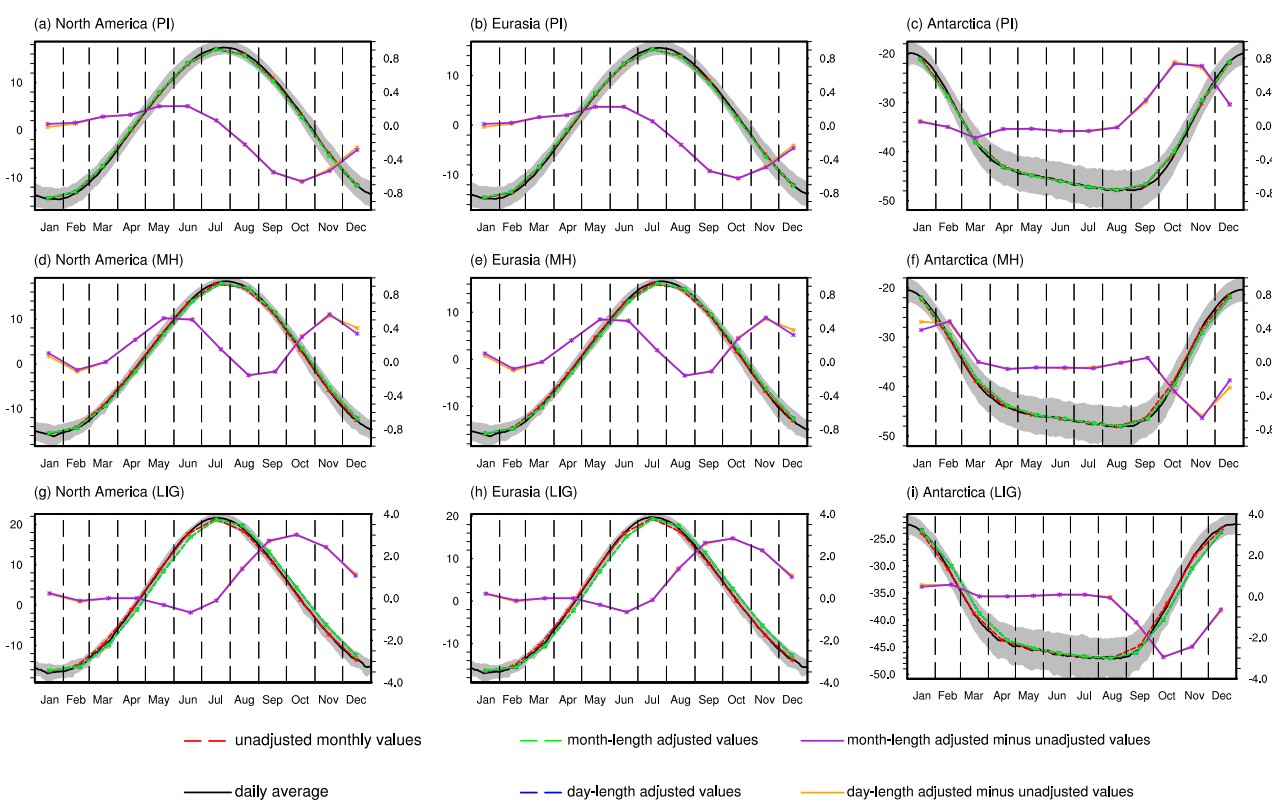

**Figure 10.** Ensemble seasonal cycle of regional mean surface air temperature in daily average (black solid lines), classical monthly means (red dashed lines), day-length adjusted means (blue dashed lines), and month-length adjusted means (green dashed lines) for (a-c) PI, (d-f) MH, and (g-i) LIG, axis to the left. Grey area represents one standard deviation from the multi-model ensemble daily mean values. Purple (orange) solid line represents the month-length (day-length) adjusted minus unadjusted values, axis to the right. The values are calculated by averaging the surface air temperatures over (a,d,g) North America, (b,e,h) Eurasia, and (c,f,i) Antarctica. Units: °C.

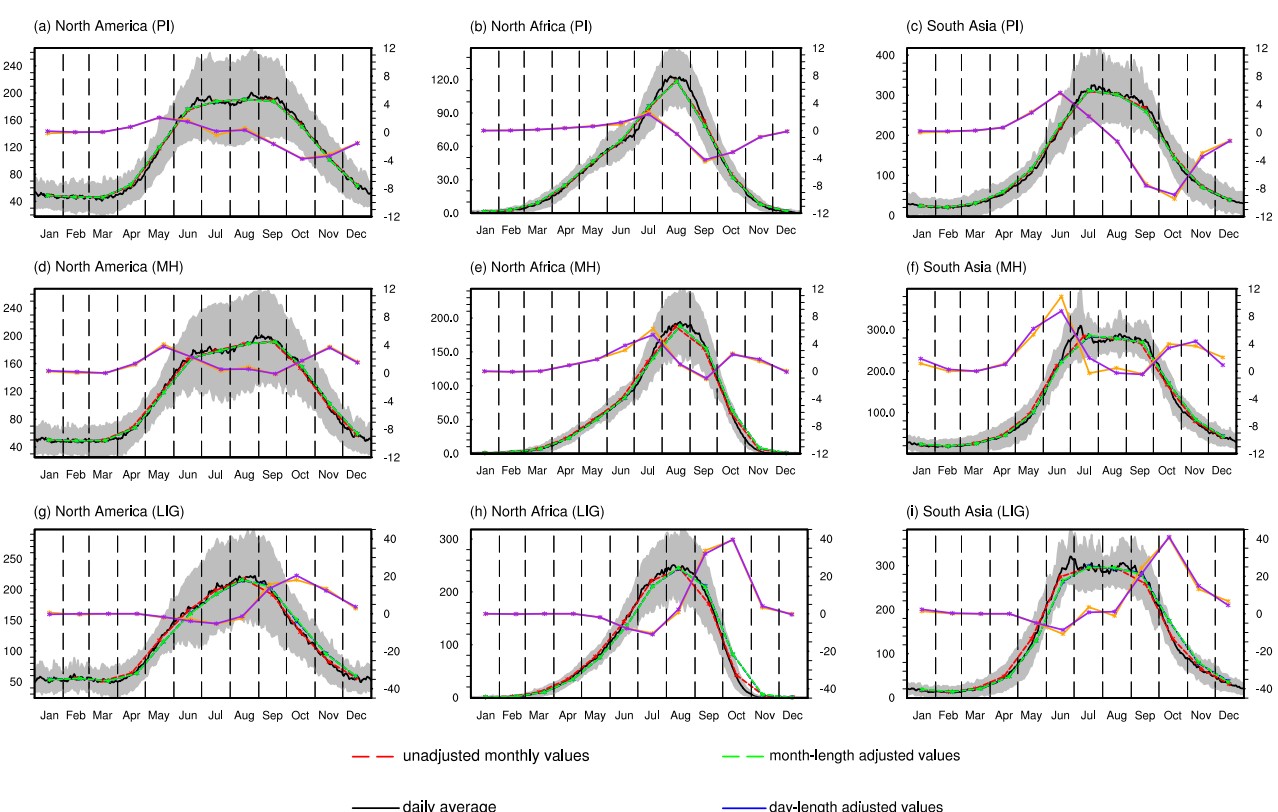

**Figure 11.** Ensemble seasonal cycle of regional mean precipitation in daily average (black solid lines), classical monthly means (red dashed lines), day-length adjusted means (blue dashed lines), and month-length adjusted means (green dashed lines) for (a-c) PI, (d-f) MH, and (g-i) LIG. Grey area represents one standard deviation from the multi-model ensemble daily mean values. Purple (orange) solid line represents the month-length (day-length) adjusted minus unadjusted values, axis to the right. The values are calculated by averaging the precipitation over (a,d,g) North America, (b,e,h) North Africa, and (c,f,i) South Asia. Units: mm/month.

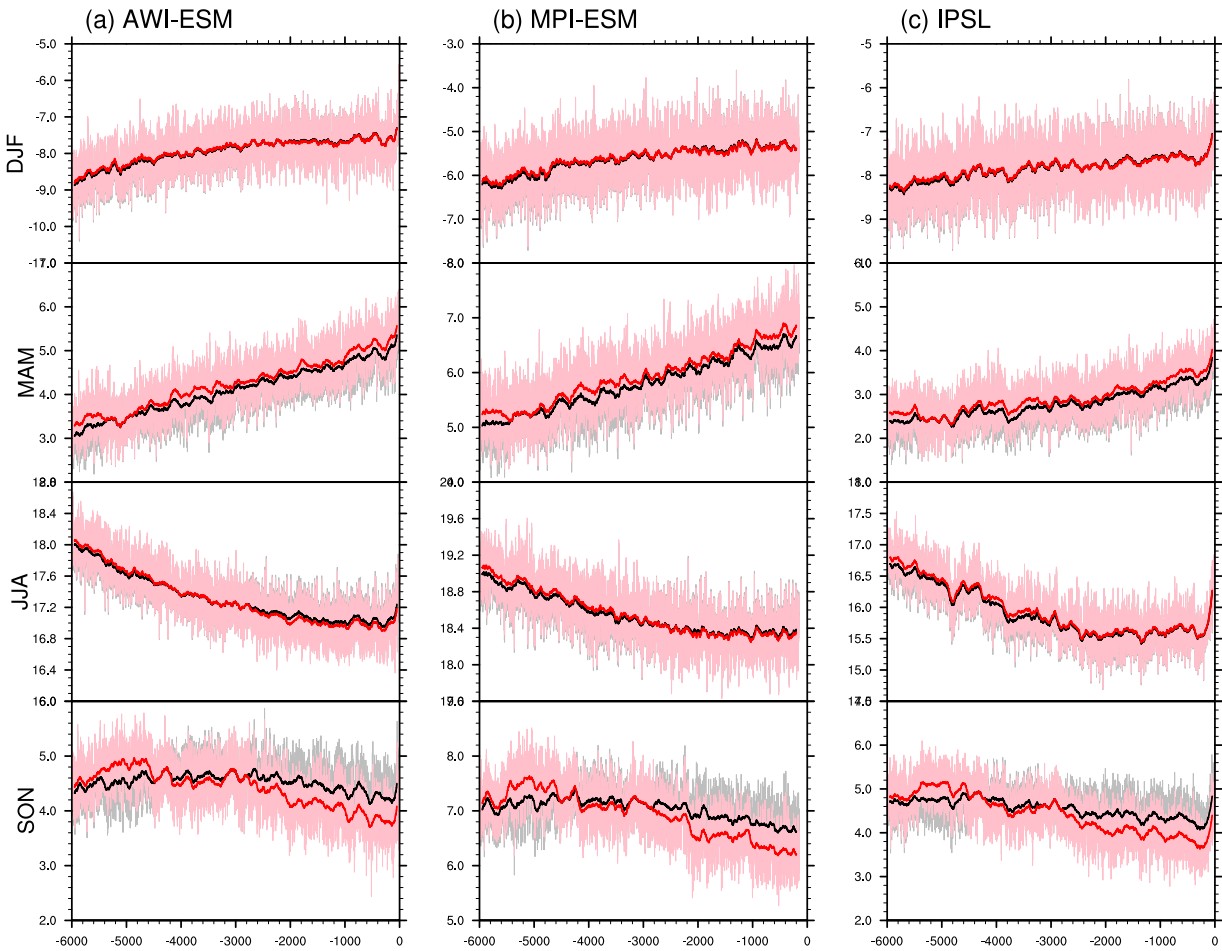

**Figure 12.** Time series of surface air temperature in classical and angular means averaged over Northern Hemisphere continents, weighted by month length, for (a) AWI-ESM, (b) MPI-ESM, and (c) IPSL-CM. Grey and pink lines stand for the original classical and angular means respectively. Smoothed curves with a running window of 100 model years are shown in black (for classical means) and red (for angular means). Units: °C.

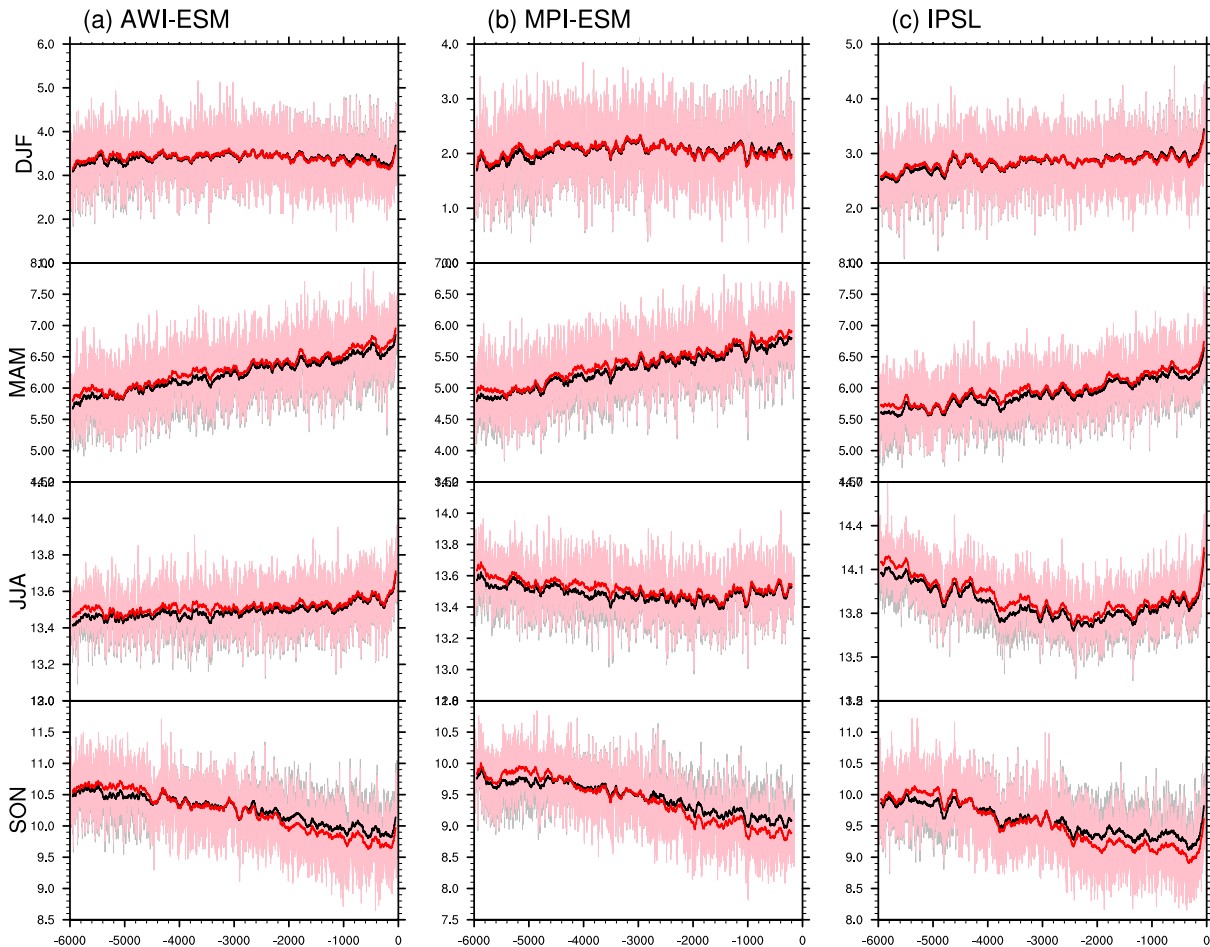

**Figure 13.** Time series of surface air temperature in classical and angular means averaged over Northern Hemisphere oceans, weighted by month length, for (a) AWI-ESM, (b) MPI-ESM, and (c) IPSL-CM. Grey and pink lines stand for the original classical and angular means respectively. Smoothed curves with a running window of 100 model years are shown in black (for classical means) and red (for angular means). Units: °C.

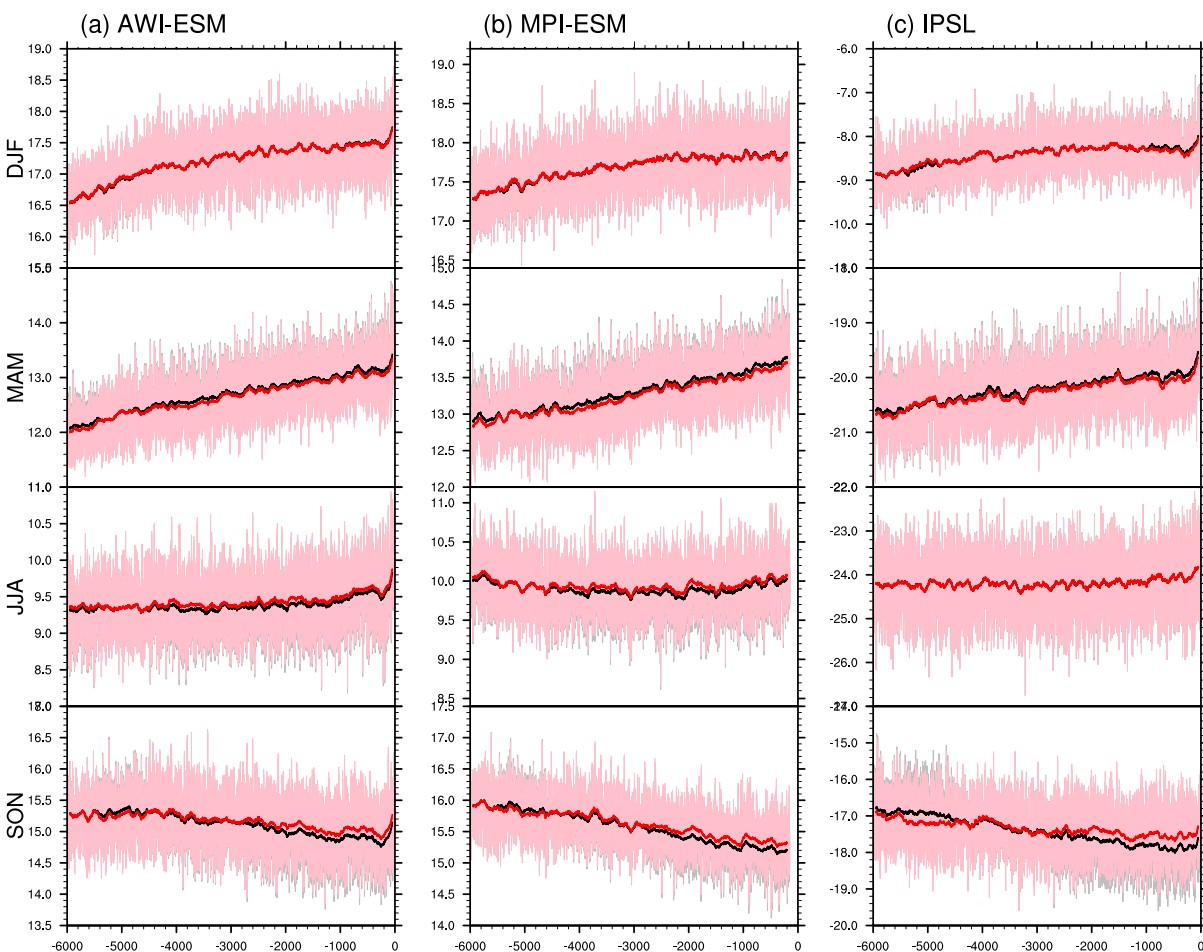

**Figure 14.** Time series of surface air temperature in classical and angular means averaged over Southern Hemisphere continents, weighted by month length, for (a) AWI-ESM, (b) MPI-ESM, and (c) IPSL-CM. Grey and pink lines stand for the original classical and angular means respectively. Smoothed curves with a running window of 100 model years are shown in black (for classical means) and red (for angular means). Units: °C.

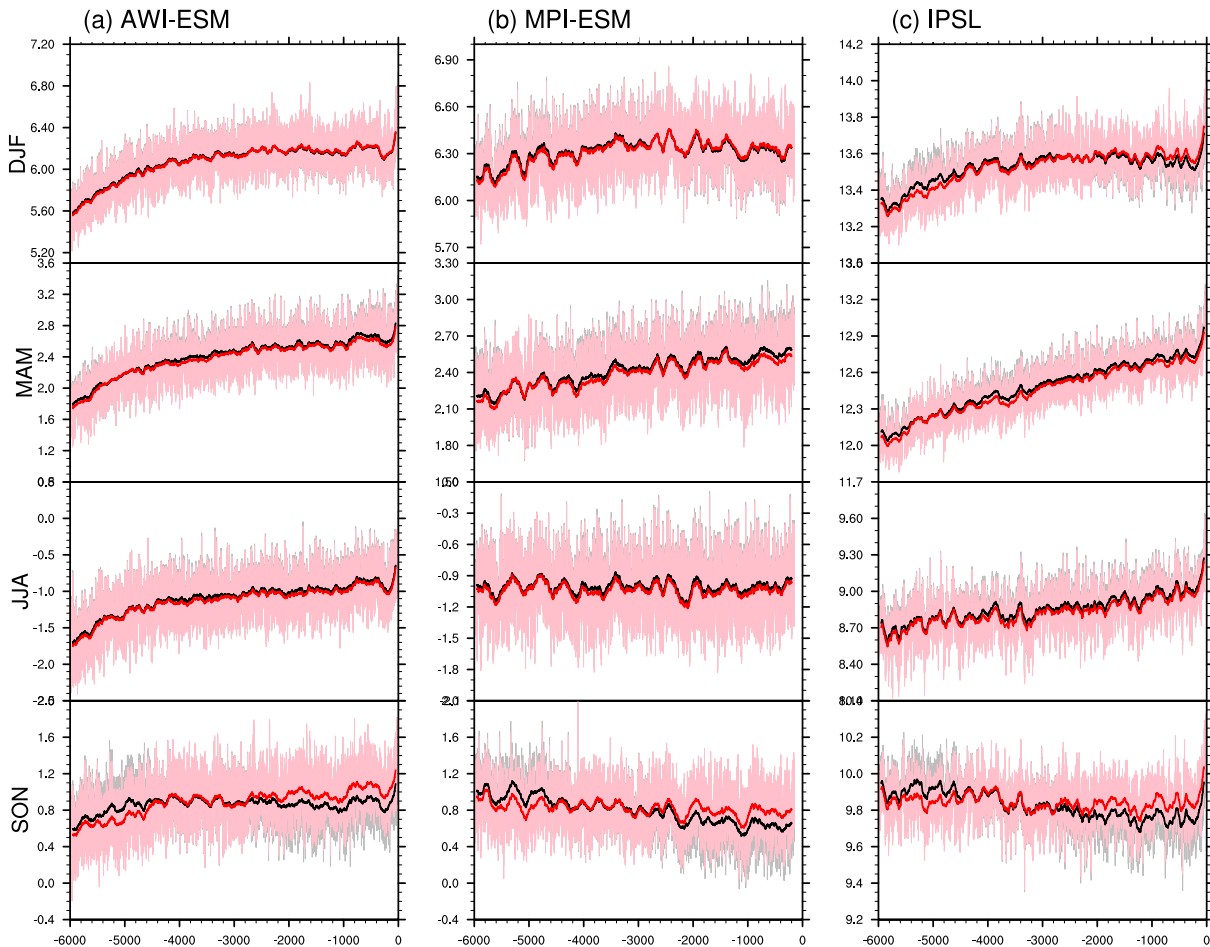

**Figure 15.** Time series of surface air temperature in classical and angular means averaged over Southern Hemisphere oceans, weighted by month length, for (a) AWI-ESM, (b) MPI-ESM, and (c) IPSL-CM. Grey and pink lines stand for the original classical and angular means respectively. Smoothed curves with a running window of 100 model years are shown in black (for classical means) and red (for angular means). Units: °C.