# Peer review of "Calendar effects on surface air temperature and precipitation based on model-ensemble equilibrium and transient simulations from PMIP4 and PACMEDY"

_Climate of the Past, 2021_

## Author Comment (AC1)

**Response letter**

Xiaoxu Shi[1], Martin Werner[1], Carolin Krug[1,2], Chris M. Brierley[3], Anni Zhao[3], Endurance Igbinosa[1,2], Pascale Braconnot[4], Esther Brady[5], Jian Cao[6], Roberta D'Agostino[7], Johann Jungclaus[7], Xingxing Liu[8], Bette Otto-Bliesner[5], Dmitry Sidorenko[1], Robert Tomas[5], Evgeny M. Volodin[9], Hu Yang[1], Qiong Zhang[10], Weipeng Zheng[11], and Gerrit Lohmann[1,2]

[1]Alfred Wegener Institute, Helmholtz Center for Polar and Marine Research, Bremerhaven, Germany
[2]Bremen University, Bremen, Germany
[3]Department of Geography, University College London, London, UK
[4]Laboratoire des Sciences du Climat et de l'Environnement-IPSL, Unité Mixte CEA-CNRS-UVSQ, Université Paris-Saclay, Orme des Merisiers, Gif-sur-Yvette, France
[5]Climate and Global Dynamics Laboratory, National Center for Atmospheric Research (NCAR), Boulder, CO 80305, USA
[6]School of Atmospheric Sciences, Nanjing University of Information Science & Technology, Nanjing, 210044, China
[7]Max Planck Institute for Meteorology, Hamburg, Germany
[8]State Key Laboratory of Loess and Quaternary Geology, Institute of Earth Environment, Chinese Academy of Sciences, Xi'an, 710061, China
[9]Marchuk Institute of Numerical Mathematics, Russian Academy of Sciences, ul. Gubkina 8, Moscow, 119333, Russia
[10]Department of Physical Geography and Bolin Centre for Climate Research, Stockholm University, 10691, Stockholm, Sweden
[11]LASG, Institute of Atmospheric Physics, Chinese Academy of Sciences, Beijing, 100029, China

The manuscript presents an analysis of PMIP4 simulations for the PI, MH and LIG and investigates the importance of the definition of the calendar. Although this has been done previously and the new results largely confirm previous ones, this new analysis is still useful as it includes an ensemble of climate model simulations and thus allows one to test the robustness of the findings over multiple models.

**Dear Reviewer,**

**Thank you very much for your positive and constructive comments. In the following, we present our point-to-point responses. Our answers to your comments are written in bold.**

**Thanks again for your time and efforts.**

**Best,**

**Xiaoxu**

Major comment:

Lines 92-93: In the literature various methods are presented to adjust monthly data towards a angular calendar. In this manuscript reference is made to Rymes and Myers (2001), but how different or similar are the various methods? So for instance Bartlein and Shafer (2019) and the various other methods that they mention in their publication (Pollard and Reusch, 2002; Timm et al., 2008; Chen et al., 2011). It would be very informative for the reader to know whether the results presented in this manuscript generally hold for all those methods or if some should be avoided.

**Thanks for the comment, we added a discussion about those approaches in the revised manuscript:**

*Various methods for adjusting monthly data towards an angular calendar have been suggested. Rymes and Myers (2001) developed a mean-preserving running-mean algorithm to reconstruct the annual cycle. In Pollard and Reusch (2002), the reconstruction of an annual cycle was based on a spline method, which fits each monthly segment by a parabola, requiring the same monthly means as the originals and continuity of value and slope at the month boundaries. Bartlein and Shafer (2019), used a mean-preserving harmonic interpolation method described in Epstein (1991) and performed the same function as the parabolic-spline interpolation method as in Pollard and Reusch (2002). To sum up, the basic procedure is similar in all the approaches, as they are all based on "mean-preserving" algorithm. In Bartlein and Shafer (2019), a comparison was made between the linear and mean-preserving interpolation methods. They found that the difference between the original monthly means and the monthly means of the linearly interpolated daily values is not negligible for both surface air temperature and precipitation while the difference between an original monthly mean value and one calculated using the mean-preserving interpolation method is negligible.*

Bartlein and Shafer (2019) made their code to perform the calendar adjustment freely available and 'user friendly'. It would be great if the same could be done with the code used in this manuscript. A reference to the code could then be added in the manuscript.

**Thanks for the suggestion, we have created a Gitlab repository and introduced it in the section of "code availability":**
*The Python source code and related manual are available from https://gitlab.awi.de/xshi/calendar (last access 02.02.2022).*

Minor comments:

Line 60: Scussolini et al. 2019 do show LIG results for precipitation and temperature for both the classical calendar and the angular calendar.

**The following texts can now be found in the revised manuscript:**
*"However, the calendar effect has been investigated in only a few paleoclimate studies. Differences of seasonal ensemble anomalies (LIG minus PI) based on the angular and the classical calendars have been shown by Scussolini et al. (2019) for both precipitation and surface air temperature. Their results indicated pronounced artificial bias for the classical calendar definition: The Northern Hemisphere warming (LIG minus PI) in boreal summer is largely underestimated. Moreover, the Northern Hemisphere monsoon precipitation during the LIG is overestimated in boreal summer but underestimated in boreal autumn. These results are in line with the findings of Joussaume and Braconnot (1997)."*

Line 72: Perhaps it is good to mention that in the results section you will first briefly describe the main features of simulated MH and LIG temperatures and precipitation (describe in more detail in previous publications) and after that you will focus on the main topic of the manuscript, namely calendar-effects.

**Thanks for the suggestion, now we changed the texts into:**
*"In the present study, we use the PMIP4 dataset to investigate the calendar effect on the simulated surface air tempera-tures and precipitation under MH and LIG boundary conditions. The structure of the paper is as follows: In Section 2, we*

*describe the method for defining an angular calendar based on the Earth's orbital parameters and provide detailed information on the data we used. In Section 3 we first briefly describe the main features of simulated MH and LIG surface air temperatures and precipitation, then we illustrate the effects of the angular season definition on the simulated patterns. We discuss and conclude in Section 4."*

Line 110: For consistency it would be better to mention the initialization procedure of all three transient simulations, not just for IPSL.

**The transient simulation by AWIESM and MPIESM is initialized from respective 1,000-year mid-Holocene spin-up run. Now we have added this information in the revised manuscript.**

Line 141: Perhaps good to not only focus on the comparison to earlier work on PMIP4 results, but also shortly on previous iterations of PMIP and other projects. For instance Lunt et al., 2013; Scussolini et al. 2019.

**Thanks for the comment, in the revised version we added the two references:**

*Our results in terms of the responses of the surface air temperature and precipitation to the MH and LIG boundary conditions are in good agreement with the results from the full PMIP4 ensemble as described in Brierley et al. (2020), Otto-Bliesner et al. (2021), and Scussolini et al. (2019), as well as the studies of earlier PMIP ensemble simulations (Lunt et al., 2013).*

Section 3.3: this section is rather long. Consider breaking it up in several sub-sections, for instance one on temperature, precipitation and one on using monthly data to calculate angular-seasons.

**Thanks for the comment, now we divided section 3.3 into three subsections: 3.3.1 for surface air temperature; 3.3.2 for precipitation; and 3.3.3 for calendar conversion based on monthly data.**

Lines 240-242: The authors say that these are 'significant' differences, but the meaning of the word significant is unclear and undefined in this context. Better to replace it.

**We agree that the meaning of 'significant' is not clear, according to the comment, we now re-phrased the contexts as:**
*We are aware of a slight artificial bias in month-length adjusted surface air temperature for LIG over the high-latitude continents in JJA, which is underestimated by 0.07 K.*

Lines 347-359: these lines are rather vague. A reference is made to major model-data mismatches that are being discussed in the literature (e.g. The Holocene temperature conundrum). So what do the results of this manuscript have to add to those discussions? Can an estimate be given on the possible magnitude of calendar effects on this model-data mismatch? Or, if not, how could this be investigated in future work? Please clarify the link between the current manuscript and the work that is mentioned in this last paragraph.

**Thanks for the comment, in the revised manuscript we have modified the paragraph and discussed about how calendar conversion impacts the model-data comparison. The new texts are as following:**

*Proxy-based reconstructions provide us another ability to examine the temperature evolution of the past and can help assess the model's performance in simulating the past climates. Since paleoclimate data often records the seasonal signal*

*(e.g. local summer temperature), an appropriate choice of calendar is therefore important for temperature comparisons between model results and proxy data. For the mid-Holocene, Bartlein et al. (2011) is an often-cited study that compiled pollen-based continental temperature reconstructions. The question arises whether the consideration of calendar effects could lead to an improved model-data agreement. Here we show in Fig. S11 the simulated classical mean temperature anomalies (MH minus PI) versus continental reconstructions. The expected increased seasonality occurs only over Northwest Europe as indicated by the proxy records. The opposite sign is shown over northern America, with winter warming and summer cooling, and is therefore not consistent with the ensemble model result. Bartlein et al. (2011) attributes such a model-data mismatch to changes in local atmospheric circulation that tend to overwhelm the insolation effect. The calendar impacts, as illustrated in Fig. 4, result in warming of less than 0.2 K over the Northern Hemisphere in both DJF and JJA, implying that model-data consistency is improved for Northwest Europe in summer, and Northern America in winter, while for most other regions using the adjusted calendar results in a poorer match between model and proxy temperatures. These results reveal that for the mid-Holocene the calendar adjustment does not guarantee a better model-data agreement, and the underlying reason might be that, in addition to the solar insolation, the proxy could be strongly influenced by the local environment, such as flow of humid air and increased cloud cover (Harrison et al., 2003) or warm-air advection (Bonfils et al., 2004).*

*Since there are very few high-resolution reconstructed temperature records for the LIG, we use here the compilation from Turney and Jones (2010) for the annual mean temperature anomalies between LIG and PI, and compare them with modeled classical mean values for boreal summer (Fig. S12). We keep in mind that the summer mean LIG temperatures are usually higher than the annual mean values documented by the proxy records. At high latitudes of Northern Hemisphere continents (e.g. Greenland, Russia and Alaska), as well as over subpolar oceans (e.g. the Nordic Sea and the Labrador Sea), we find that the models underestimate the recorded LIG warming. Part of the bias can be corrected by calendar adjustment which leads to a warming of up to 1 K over Northern Hemisphere continents in JJA (Fig. 3k).*

Figures 5-7: It is always a difficult choice whether to show precipitation changes in units of mm/time or as percentages. The authors choose to use mm/month and as a result the tropical regions supposedly show the most marked changes in precipitation while in terms of percentages the picture might look quite different. Consider adding figures to the supplement that show percentage precipitation changes.

**Thanks for the suggestion. It is a good idea to examine the calendar effect on precipitation with both the absolute changes and percentage changes. In the revised version, we have added a new supplementary figure to show the percentage changes of precipitation (see Fig. S7 in the new manuscript).**

**We also updated the related texts:**

*In LIG, the largest calendar effects on precipitation can be observed for SON over the tropical rain-belt (Fig. 6 shows the anomalies and Fig. S7 shows the percentage changes), with positive anomalies (within 30 mm/month) to the north and negative anomalies (up to -30 mm/month) to the south of the Inter Tropical Convergence Zone (ITCZ). In North Africa, changes in precipitation due to calendar transition account for up to 80% of the classical mean (Fig. S7d). In DJF, we*

*observe a tripole pattern, with negative anomalies over North (-1 mm/month, -10%) and South Africa (-4 mm/month, -5%) and positive anomalies over equatorial Africa (5 mm/month, 8%). For JJA the adjusted-minus-unadjusted precipitation anomalies present a dryness (up to -15 mm/month, -15%) and wetness (less than 10 mm/month, 16%) over the northern*
120 *and southern edge of the ITCZ, respectively, opposite to the patterns for SON and DJF.*

Technical comments:

Line 44: replace the word 'bunch'

**We changed it into *"a number of modelling groups"***

Lines 57-58: "hereafter referred to as fixed-length or classical calendar". Perhaps better to use only one of the two in the
125 remainder of the manuscript to avoid confusion.

**We now used "classical calendar" throughout the manuscript.**

Lines 107, 111: use subscripts for the names of the greenhouse-gasses.

**We now changed the names for the greenhouse gases into "$CO_2$, $CH_4$ and $N_2O$ "**

Line 130: replace 'at the' by 'over' or perhaps 'in'?
130 **Thanks for the correction. We changed "at the" into "over".**

Main article figures and supplementary figures: Just for clarity, mention in the figure captions when the figure shows multi-model-mean results.

**Thanks for the suggestion. We have now indicated in the captions when the results are ensemble.**

**References**

135  Bartlein, P. J. and Shafer, S. L.: Paleo calendar-effect adjustments in time-slice and transient climate-model simulations (PaleoCalAdjust v1.
     0): Impact and strategies for data analysis, Geoscientific Model Development, 12, 3889–3913, 2019.

Bartlein, P. J., Harrison, S., Brewer, S., Connor, S., Davis, B., Gajewski, K., Guiot, J., Harrison-Prentice, T., Henderson, A., Peyron, O., et al.:
     Pollen-based continental climate reconstructions at 6 and 21 ka: a global synthesis, Climate Dynamics, 37, 775–802, 2011.

Bonfils, C., de Noblet-Ducoudré, N., Guiot, J., and Bartlein, P.: Some mechanisms of mid-Holocene climate change in Europe, inferred from
140  comparing PMIP models to data, Climate Dynamics, 23, 79–98, 2004.

Brierley, C. M., Zhao, A., Harrison, S. P., Braconnot, P., Williams, C. J., Thornalley, D. J., Shi, X., Peterschmitt, J.-Y., Ohgaito, R., Kaufman,
     D. S., et al.: Large-scale features and evaluation of the PMIP4-CMIP6 midHolocene simulations, Climate of the Past, 16, 1847–1872,
     2020.

Epstein, E. S.: On obtaining daily climatological values from monthly means, Journal of Climate, 4, 365–368, 1991.

145  Harrison, S. P. a., Kutzbach, J. E., Liu, Z., Bartlein, P. J., Otto-Bliesner, B., Muhs, D., Prentice, I. C., and Thompson, R. S.: Mid-Holocene
     climates of the Americas: a dynamical response to changed seasonality, Climate Dynamics, 20, 663–688, 2003.

Joussaume, S. and Braconnot, P.: Sensitivity of paleoclimate simulation results to season definitions, Journal of Geophysical Research:
     Atmospheres, 102, 1943–1956, 1997.

Lunt, D., Abe-Ouchi, A., Bakker, P., Berger, A., Braconnot, P., Charbit, S., Fischer, N., Herold, N., Jungclaus, J. H., Khon, V., et al.: A
150  multi-model assessment of last interglacial temperatures, Climate of the Past, 9, 699–717, 2013.

Otto-Bliesner, B. L., Brady, E. C., Zhao, A., Brierley, C. M., Axford, Y., Capron, E., Govin, A., Hoffman, J. S., Isaacs, E., Kageyama, M.,
     et al.: Large-scale features of Last Interglacial climate: results from evaluating the lig127k simulations for the Coupled Model Intercom-
     parison Project (CMIP6)–Paleoclimate Modeling Intercomparison Project (PMIP4), Climate of the Past, 17, 63–94, 2021.

Pollard, D. and Reusch, D. B.: A calendar conversion method for monthly mean paleoclimate model output with orbital forcing, Journal of
155  Geophysical Research: Atmospheres, 107, ACL–3, 2002.

Rymes, M. and Myers, D.: Mean preserving algorithm for smoothly interpolating averaged data, Solar Energy, 71, 225–231, 2001.

Scussolini, P., Bakker, P., Guo, C., Stepanek, C., Zhang, Q., Braconnot, P., Cao, J., Guarino, M.-V., Coumou, D., Prange, M., et al.: Agreement
     between reconstructed and modeled boreal precipitation of the Last Interglacial, Science advances, 5, eaax7047, 2019.

Turney, C. S. and Jones, R. T.: Does the Agulhas Current amplify global temperatures during super-interglacials?, Journal of Quaternary
160  Science, 25, 839–843, 2010.

---

## Author Comment (AC2)

**Response letter**

Xiaoxu Shi1, Martin Werner1, Carolin Krug1,2, Chris M. Brierley3, Anni Zhao3, Endurance Igbinosa1,2, Pascale Braconnot4, Esther Brady5, Jian Cao6, Roberta D'Agostino7, Johann Jungclaus7, Xingxing Liu8, Bette Otto-Bliesner5, Dmitry Sidorenko1, Robert Tomas5, Evgeny M. Volodin9, Hu Yang1, Oiong Zhang10, Weipeng Zheng11, and Gerrit Lohmann1,2 1Alfred Wegener Institute, Helmholtz Center for Polar and Marine Research, Bremerhaven, Germany 2Bremen University, Bremen, Germany 3Department of Geography, University College London, London, UK 4Laboratoire des Sciences du Climat et de l'Environnement-IPSL, Unité Mixte CEA-CNRS-UVSO, Université Paris-Saclay, Orme des Merisiers, Gif-sur-Yvette, France 5Climate and Global Dynamics Laboratory, National Center for Atmospheric Research (NCAR), Boulder, CO 80305, USA 6School of Atmospheric Sciences, Nanjing University of Information Science & Technology, Nanjing, 210044, China 7Max Planck Institute for Meteorology, Hamburg, Germany 8State Key Laboratory of Loess and Quaternary Geology, Institute of Earth Environment, Chinese Academy of Sciences, Xi'an, 710061, China 9Marchuk Institute of Numerical Mathematics, Russian Academy of Sciences, ul. Gubkina 8, Moscow, 119333, Russia 10Department of Physical Geography and Bolin Centre for Climate Research, Stockholm University, 10691, Stockholm, Sweden

11LASG, Institute of Atmospheric Physics, Chinese Academy of Sciences, Beijing, 100029, China

**1 Comments from Reviewer 2**

Calendar effect is a problem in paleoclimate modelling since long. To my knowledge, there are not many studies dedicating to this topic. In this manuscript, the authors investigate the calendar effect on seasonal temperature and precipitation by using the PMIP simulations of the PI, 6ka, 127ka snapshot and the 6-0ka transient experiments. The results could be informative for

5 paleoclimate community and let pay more attention to the calendar problem. In the meantime, some improvements would be needed for clarifications and also to make the manuscript more attractive. Please find my comments and questions here below.

**Dear Reviewer,**

Thank you very much for your positive and constructive comments. In the following, we present our point-to-point responses. Our answers to your comments are written in **bold**.

10 Thanks again for your time and efforts.

Best,

Xiaoxu

General comments:

1. There are not many studies on the calendar effect in paleoclimate simulations. In addition to Joussaume and Braconnot

15 1997, Bartlein and Shafer 2019 cited in the manuscript, there are also Pollard and Reusch 2002 (https://doi.org/10.1029/2002JD002126

), Timm et al 2008 (https://doi.org/10.1029/2007PA001461 ) and Chen et al 2010 (https://doi.org/10.1007/s00382-010-0944-6)

. These studies and their findings should be mentioned in the manuscript and be compared with.

Thanks for the suggestion. Now in our revised manuscript we added the following texts into the discussion part: In previous studies, the angular calendar was defined using the true anomaly of the Earth corresponding to the present-

- 20 day seasons, in other words, each month begins and ends at the same celestial longitude as present-day for any period (Joussaume and Braconnot, 1997; Bartlein and Shafer, 2019; Timm et al., 2008; Chen et al., 2011; Pollard and Reusch, 2002). The work of Chen et al. (2011) and Timm et al. (2008) applied a 360-day year which is, originally, divided into 12 months with 30 days. The VE is set to day 81 in a calendar year. Pollard and Reusch (2002), Joussaume and Braconnot (1997) and Bartlein and Shafer (2019), on the other hand, performed the calendar adjustment based on today's classical
- 25 calendar with 365 days in a non-leap year. In their studies, an assumption was made that the seasonality defined by the classical calendar is in phase with the insolation and solar geometry for modern-day. In our study, by calculating the onset of present-day months/seasons using the approach described in Section 2.1, we find that the classical "fixed-length" calendar is very similar to the angular calendar for today, but they are not completely the same. This is evidenced in the small shift of months between the two calendars as seen in Table 3. In particular the angular October is delayed by 3 days com-
- 30 pared to the classical October, resulting in negative anomalies in the adjusted-minus-unadjusted solar insolation. Though different methods are used in our work from the mentioned previous studies, our results are identical: for the LIG, the adjusted-minus-unadjusted surface air temperature over the Northern Hemisphere is up to 5 K during SON (Joussaume and Braconnot, 1997; Bartlein and Shafer, 2019; Chen et al., 2011) or September (Pollard and Reusch, 2002); and the Northern Hemisphere monsoon precipitation in SON is underestimated by the use of the classical calendar (Bartlein and
- 35 Shafer, 2019; Chen et al., 2011). Similar biases are found for the early-Holocene (Timm et al., 2008) and mid-Holocene (Joussaume and Braconnot, 1997; Bartlein and Shafer, 2019) but less pronounced. These results are consistent with the findings in our study, however, comparing results of our 3 transient simulations with that from the TraCE-21ka transient simulation, as it was investigated in Bartlein and Shafer (2019), distinct differences emerge for the boreal autumn surface air temperature near present-day. In Bartlein and Shafer (2019), the artificial bias in MH-minus-PI temperature and pre-
- 40 cipitation totally stems from the bias in MH when the classical calendar is applied (as for PI both calendars are identical). In contrast, our study reveals that such bias is mainly dominated by the deviation between angular and classical calendars for present-day. It should be noted that these discrepancies are not due to the different models used in our studies, but rather to the different approaches adopted for calendar adjustment.

It is unclear for me how the conversion of temperature and precipitation on the classical calendar to those on angular
 calendar was made. A thorough explanation would be needed in section 2.1. Please also see some of my specific comments.

**We totally agree to the reviewer that we should give more detailed description on the calendar conversion method. In the revised paper we have done so, please see the updated section 2.1, as well as our responses to your specific comments.**

3. The calendar effect at 6 ka and 127 ka was examined using multi-model ensemble. Is the calendar effect on temperature and precipitation similar between individual models qualitatively and quantitatively speaking? Additional analysis on individual

**Figure 1.** (a,b) Deviation of MH-PI SON surface air temperature between angular and classical means for (a) continents and (b) oceans at different latitude-bands, simulated by individual models. (c,d) As in (a,b), but for LIG-PI surface air temperature. Units: K.

50 model would be interesting. Moreover, how is the calendar effect compared to the difference between models? For example, the difference between the black and pink lines in Fig.12 appears very small. It is much smaller than the model-model difference. Is such a small effect worth to be mentioned?

Thanks for the constructive comment, in the revised paper we have added two plots for such an analysis (Fig. 1 and Fig. 2 in this letter). We also put our original Fig. 12 to the supplementary. Here are the related plots and texts for the calendar effect on temperature and precipitation between individual models (here we take SON as it has the largest calendar effect and is therefore more interesting than the other seasons):

3

---

## Author Response (AR1)

**Response letter to Reviewer 1**

Xiaoxu Shi[1], Martin Werner[1], Carolin Krug[1,2], Chris M. Brierley[3], Anni Zhao[3], Endurance Igbinosa[1,2], Pascale Braconnot[4], Esther Brady[5], Jian Cao[6], Roberta D'Agostino[7], Johann Jungclaus[7], Xingxing Liu[8], Bette Otto-Bliesner[5], Dmitry Sidorenko[1], Robert Tomas[5], Evgeny M. Volodin[9], Hu Yang[1], Qiong Zhang[10], Weipeng Zheng[11], and Gerrit Lohmann[1,2]

[1]Alfred Wegener Institute, Helmholtz Center for Polar and Marine Research, Bremerhaven, Germany
[2]Bremen University, Bremen, Germany
[3]Department of Geography, University College London, London, UK
[4]Laboratoire des Sciences du Climat et de l'Environnement-IPSL, Unité Mixte CEA-CNRS-UVSQ, Université Paris-Saclay, Orme des Merisiers, Gif-sur-Yvette, France
[5]Climate and Global Dynamics Laboratory, National Center for Atmospheric Research (NCAR), Boulder, CO 80305, USA
[6]School of Atmospheric Sciences, Nanjing University of Information Science & Technology, Nanjing, 210044, China
[7]Max Planck Institute for Meteorology, Hamburg, Germany
[8]State Key Laboratory of Loess and Quaternary Geology, Institute of Earth Environment, Chinese Academy of Sciences, Xi'an, 710061, China
[9]Marchuk Institute of Numerical Mathematics, Russian Academy of Sciences, ul. Gubkina 8, Moscow, 119333, Russia
[10]Department of Physical Geography and Bolin Centre for Climate Research, Stockholm University, 10691, Stockholm, Sweden
[11]LASG, Institute of Atmospheric Physics, Chinese Academy of Sciences, Beijing, 100029, China

The manuscript presents an analysis of PMIP4 simulations for the PI, MH and LIG and investigates the importance of the definition of the calendar. Although this has been done previously and the new results largely confirm previous ones, this new analysis is still useful as it includes an ensemble of climate model simulations and thus allows one to test the robustness of the findings over multiple models.

**Dear Reviewer,**

**Thank you very much for your positive and constructive comments. In the following, we present our point-to-point responses. Our answers to your comments are written in bold.**

**Thanks again for your time and efforts.**

**Best,**

**Xiaoxu**

Major comment:

Lines 92-93: In the literature various methods are presented to adjust monthly data towards a angular calendar. In this manuscript reference is made to Rymes and Myers (2001), but how different or similar are the various methods? So for instance Bartlein and Shafer (2019) and the various other methods that they mention in their publication (Pollard and Reusch, 2002; Timm et al., 2008; Chen et al., 2011). It would be very informative for the reader to know whether the results presented in this manuscript generally hold for all those methods or if some should be avoided.

**Thanks for the comment, we added a discussion about those approaches in the revised manuscript (please also see L394-404 in the manuscript and L421-433 in the difference-tracked version):**

*Various methods for adjusting monthly data towards an angular calendar have been suggested. Rymes and Myers (2001)*
*developed a mean-preserving running-mean algorithm to reconstruct the annual cycle. In Pollard and Reusch (2002),*
*the reconstruction of an annual cycle was based on a spline method, which fits each monthly segment by a parabola,*
*requiring the same monthly means as the originals and continuity of value and slope at the month boundaries. Bartlein and*
*Shafer (2019), used a mean-preserving harmonic interpolation method described in Epstein (1991) and performed the same*
*function as the parabolic-spline interpolation method as in Pollard and Reusch (2002). To sum up, the basic procedure*
*is similar in all the approaches, as they are all based on "mean-preserving" algorithm. In Bartlein and Shafer (2019),*
*a comparison was made between the linear and mean-preserving interpolation methods. They found that the difference*
*between the original monthly means and the monthly means of the linearly interpolated daily values is not negligible*
*for both surface air temperature and precipitation while the difference between an original monthly mean value and one*
*calculated using the mean-preserving interpolation method is negligible.*

Bartlein and Shafer (2019) made their code to perform the calendar adjustment freely available and 'user friendly'. It would be great if the same could be done with the code used in this manuscript. A reference to the code could then be added in the manuscript.

**Thanks for the suggestion, we have created a Gitlab repository and introduced it in the section of "code availability":**
*The Python source code and related manual are available from https://gitlab.awi.de/xshi/calendar (last access 02.02.2022).*

Minor comments:

Line 60: Scussolini et al. 2019 do show LIG results for precipitation and temperature for both the classical calendar and the angular calendar.

**The following texts can now be found at L64-73 of the revised manuscript as well as L72-78 of the difference-tracked version:**

*"However, the calendar effect has been investigated in only a few paleoclimate studies. Differences of seasonal ensemble*
*anomalies (LIG minus PI) based on the angular and the classical calendars have been shown by Scussolini et al. (2019) for*
*both precipitation and surface air temperature. Their results indicated pronounced artificial bias for the classical calendar*
*definition: The Northern Hemisphere warming (LIG minus PI) in boreal summer is largely underestimated. Moreover,*
*the Northern Hemisphere monsoon precipitation during the LIG is overestimated in boreal summer but underestimated in*
*boreal autumn. These results are in line with the findings of Joussaume and Braconnot (1997)."*

Line 72: Perhaps it is good to mention that in the results section you will first briefly describe the main features of simulated MH and LIG temperatures and precipitation (describe in more detail in previous publications) and after that you will focus on the main topic of the manuscript, namely calendar-effects.

**Thanks for the suggestion, now we changed the texts into:**

*"In the present study, we use the PMIP4 dataset to investigate the calendar effect on the simulated surface air temperatures and precipitation under MH and LIG boundary conditions. The structure of the paper is as follows: In Section 2, we describe the method for defining an angular calendar based on the Earth's orbital parameters and provide detailed information on the data we used. In Section 3 we first briefly describe the main features of simulated MH and LIG surface air temperatures and precipitation, then we illustrate the effects of the angular season definition on the simulated patterns. We discuss and conclude in Section 4 and 5, respectively."*

**The above texts could be found at L83-88 in the revised manuscript and L89-94 in the difference-tracked version.**

Line 110: For consistency it would be better to mention the initialization procedure of all three transient simulations, not just for IPSL.

**The transient simulation by AWIESM and MPIESM is initialized from respective 1,000-year mid-Holocene spin-up run. Now we have added this information in the revised manuscript. We refer to section 2.2 in our manuscript.**

Line 141: Perhaps good to not only focus on the comparison to earlier work on PMIP4 results, but also shortly on previous iterations of PMIP and other projects. For instance Lunt et al., 2013; Scussolini et al. 2019.

**Thanks for the comment, in the revised version we added the two references:**

*Our results in terms of the responses of the surface air temperature and precipitation to the MH and LIG boundary conditions are in good agreement with the results from the full PMIP4 ensemble as described in Brierley et al. (2020), Otto-Bliesner et al. (2021), and Scussolini et al. (2019), as well as the studies of earlier PMIP ensemble simulations (Lunt et al., 2013).*

**The above texts could be found at L192-195 in the revised manuscript and L201-204 in the difference-tracked version.**

Section 3.3: this section is rather long. Consider breaking it up in several sub-sections, for instance one on temperature, precipitation and one on using monthly data to calculate angular-seasons.

**Thanks for the comment, now we divided section 3.3 into three subsections: 3.3.1 for surface air temperature; 3.3.2 for precipitation; and 3.3.3 for calendar conversion based on monthly data.**

Lines 240-242: The authors say that these are 'significant' differences, but the meaning of the word significant is unclear and undefined in this context. Better to replace it.

**We agree that the meaning of 'significant' is not clear, according to the comment, we now re-phrased the contexts as:**
*We are aware of a slight artificial bias in month-length adjusted surface air temperature for LIG over the high-latitude continents in JJA, which is underestimated by 0.07 K.*

Lines 347-359: these lines are rather vague. A reference is made to major model-data mismatches that are being discussed in the literature (e.g. The Holocene temperature conundrum). So what do the results of this manuscript have to add to those discussions? Can an estimate be given on the possible magnitude of calendar effects on this model-data mismatch? Or, if not, how could this be investigated in future work? Please clarify the link between the current manuscript and the work that is mentioned in this last paragraph.

**Thanks for the comment, in the revised manuscript we have modified the paragraph and discussed about how calendar conversion impacts the model-data comparison. The new texts are as following (please see L436-458 of the revised manuscript and also L484-507 of the difference-tracked version):**

*Proxy-based reconstructions provide us another ability to examine the temperature evolution of the past and can help assess the model's performance in simulating the past climates. Since paleoclimate data often records the seasonal signal (e.g. local summer temperature), an appropriate choice of calendar is therefore important for temperature comparisons between model results and proxy data. For the mid-Holocene, Bartlein et al. (2011) is an often-cited study that compiled pollen-based continental temperature reconstructions. The question arises whether the consideration of calendar effects could lead to an improved model-data agreement. Here we show in Fig. S11 the simulated classical mean temperature anomalies (MH minus PI) versus continental reconstructions. The expected increased seasonality occurs only over Northwest Europe as indicated by the proxy records. The opposite sign is shown over northern America, with winter warming and summer cooling, and is therefore not consistent with the ensemble model result. Bartlein et al. (2011) attributes such a model-data mismatch to changes in local atmospheric circulation that tend to overwhelm the insolation effect. The calendar impacts, as illustrated in Fig. 4, result in warming of less than 0.2 K over the Northern Hemisphere in both DJF and JJA, implying that model-data consistency is improved for Northwest Europe in summer, and Northern America in winter, while for most other regions using the adjusted calendar results in a poorer match between model and proxy temperatures. These results reveal that for the mid-Holocene the calendar adjustment does not guarantee a better model-data agreement, and the underlying reason might be that, in addition to the solar insolation, the proxy could be strongly influenced by the local environment, such as flow of humid air and increased cloud cover (Harrison et al., 2003) or warm-air advection (Bonfils et al., 2004).*

*Since there are very few high-resolution reconstructed temperature records for the LIG, we use here the compilation from Turney and Jones (2010) for the annual mean temperature anomalies between LIG and PI, and compare them with modeled classical mean values for boreal summer (Fig. S12). We keep in mind that the summer mean LIG temperatures are usually higher than the annual mean values documented by the proxy records. At high latitudes of Northern Hemisphere continents (e.g. Greenland, Russia and Alaska), as well as over subpolar oceans (e.g. the Nordic Sea and the Labrador Sea), we find that the models underestimate the recorded LIG warming. Part of the bias can be corrected by calendar adjustment which leads to a warming of up to 1 K over Northern Hemisphere continents in JJA (Fig. 3k).*

Figures 5-7: It is always a difficult choice whether to show precipitation changes in units of mm/time or as percentages. The authors choose to use mm/month and as a result the tropical regions supposedly show the most marked changes in precipitation while in terms of percentages the picture might look quite different. Consider adding figures to the supplement that show percentage precipitation changes.

**Thanks for the suggestion. It is a good idea to examine the calendar effect on precipitation with both the absolute changes and percentage changes. In the revised version, we have added a new supplementary figure to show the percentage changes of precipitation (see Fig. S7 in the new manuscript).**

**We also updated the related texts (please see L272-279 of the revised manuscript and also L281-289 of the difference-tracked version):**

*In LIG, the largest calendar effects on precipitation can be observed for SON over the tropical rain-belt (Fig. 6 shows the anomalies and Fig. S7 shows the percentage changes), with positive anomalies (within 30 mm/month) to the north and negative anomalies (up to -30 mm/month) to the south of the Inter Tropical Convergence Zone (ITCZ). In North Africa, changes in precipitation due to calendar transition account for up to 80% of the classical mean (Fig. S7d). In DJF, we observe a tripole pattern, with negative anomalies over North (-1 mm/month, -10%) and South Africa (-4 mm/month, -5%) and positive anomalies over equatorial Africa (5 mm/month, 8%). For JJA the adjusted-minus-unadjusted precipitation anomalies present a dryness (up to -15 mm/month, -15%) and wetness (less than 10 mm/month, 16%) over the northern and southern edge of the ITCZ, respectively, opposite to the patterns for SON and DJF.*

Technical comments:

Line 44: replace the word 'bunch'

**We changed it into *"a number of modelling groups"***

Lines 57-58: "hereafter referred to as fixed-length or classical calendar". Perhaps better to use only one of the two in the remainder of the manuscript to avoid confusion.

**We now used "classical calendar" throughout the manuscript.**

Lines 107, 111: use subscripts for the names of the greenhouse-gasses.

**We now changed the names for the greenhouse gases into "$CO_2$, $CH_4$ and $N_2O$ "**

Line 130: replace 'at the' by 'over' or perhaps 'in'?

**Thanks for the correction. We changed "at the" into "over".**

Main article figures and supplementary figures: Just for clarity, mention in the figure captions when the figure shows multi-model-mean results.

**Thanks for the suggestion. We have now indicated in the captions when the results are ensemble.**

[Figure]

**Figure 1.** (a,b) Deviation of MH-PI SON surface air temperature between angular and classical means for (a) continents and (b) oceans at different latitude-bands, simulated by individual models. (c,d) As in (a,b), but for LIG-PI surface air temperature. Units: K.

difference between the black and pink lines in Fig.12 appears very small. It is much smaller than the model-model difference. Is such a small effect worth to be mentioned?

**Thanks for the constructive comment, in the revised paper we have added two plots for such an analysis (Fig. 1 and Fig. 2 in this letter). We also put our original Fig. 12 to the supplementary. Here are the related plots and texts for the calendar effect on temperature and precipitation between individual models (here we take SON as it has the largest calendar effect and is therefore more interesting than the other seasons):**

[Figure]

**Figure 2.** (a) Deviation of MH-PI SON precipitation between angular and classical means for North America, North Africa and South Asia, simulated by individual models. (b) As in (a), but for LIG-PI precipitation. Units: mm/month.

**The following 2 paragraphs could be found at L268-270, L277-279 of the revised manuscript and also at L294-299, L304-309 of the difference-tracked version**

*Analysis on individual models reveal a robust calendar effect on the SON surface air temperature for both continents and oceans, which overwhelms the differences between models (Fig. 5). We also observe that the calendar effect on temperature anomalies is more pronounced at higher latitudes than at lower latitudes.*

*Fig. 9 depicts the calendar impact on the SON precipitation anomaly over the main monsoon domains of the Northern Hemisphere (i.e. North America, North Africa and South Asia). We notice a very large model-model discrepancy for all regions examined in both the MH and the LIG, with the exception of North Africa in the MH. Our results indicate that that during the MH, the precipitation in South Asia is more responsive to a calendar adjustment compared to North Africa and North America. However, for the LIG, no robust conclusion could be drawn about the calendar effects in the different regions due to the large discrepancies between the models.*

4. To which extent is the model-proxy comparison improved when calendar effect is considered? It would be interesting to show the comparison with proxy data before and after the calendar conversion.

**It is a good idea to discuss about the model-proxy comparison, based on the comment, we added in the discussion section (please see L436-458 of the revised manuscript and also L484-507 of the difference-tracked version):**

*Proxy-based reconstructions provide us another ability to examine the temperature evolution of the past and can help assess the model's performance in simulating the past climates. Since paleoclimate data often records the seasonal signal (e.g. local summer temperature), an appropriate choice of calendar is therefore important for temperature comparisons between model results and proxy data. For the mid-Holocene, Bartlein et al. (2011) is an often-cited study that compiled pollen-based continental temperature reconstructions. The question arises whether the consideration of calendar effects could lead to an improved model-data agreement. Here we show in Fig. S11 the simulated classical mean temperature anomalies (MH minus PI) versus continental reconstructions. The expected increased seasonality occurs only over Northwest Europe as indicated by the proxy records. The opposite sign is shown over northern America, with winter warming and summer cooling, and is therefore not consistent with the ensemble model result. Bartlein et al. (2011) attributes such a model-data mismatch to changes in local atmospheric circulation that tend to overwhelm the insolation effect. The calendar impacts, as illustrated in Fig. 4, result in warming of less than 0.2 K over the Northern Hemisphere in both DJF and JJA, implying that model-data consistency is improved for Northwest Europe in summer, and Northern America in winter, while for most other regions using the adjusted calendar results in a poorer match between model and proxy temperatures. These results reveal that for the mid-Holocene the calendar adjustment does not guarantee a better model-data agreement, and the underlying reason might be that, in addition to the solar insolation, the proxy could be strongly influenced by the local environment, such as flow of humid air and increased cloud cover (Harrison et al., 2003) or warm-air advection (Bonfils et al., 2004).*

*Since there are very few high-resolution reconstructed temperature records for the LIG, we use here the compilation from Turney and Jones (2010) for the annual mean temperature anomalies between LIG and PI, and compare them with*

*modeled classical mean values for boreal summer (Fig. S12). We keep in mind that the summer mean LIG temperatures are usually higher than the annual mean values documented by the proxy records. At high latitudes of Northern Hemisphere continents (e.g. Greenland, Russia and Alaska), as well as over subpolar oceans (e.g. the Nordic Sea and the Labrador Sea), we find that the models underestimate the recorded LIG warming. Part of the bias can be corrected by calendar adjustment which leads to a warming of up to 1 K over Northern Hemisphere continents in JJA (Fig. 3k).*

5. If I understand well, the calendar effect happens mainly when the seasonal climate between two time slices is compared. Is it worth to consider it when we are interested only in the absolute climate at one given time slice? Does the calendar effect have an influence on the simulated vegetation?

**Yes, for example Fig. 1 in the paper illustrates the calendar effect on absolute values of temperature. This is important when comparing reconstructed proxy records with model simulations for the LIG and MH. In order to answer the second question in this comment, in the supplementary we add one plot about the calendar effect on leaf area index, which shows that the lead area index is only slightly affected by the calendar definition. In the discussion section, we add the following (L459-464 of the revised manuscript and also L508-513 of the difference-tracked version):**

*Not all types of archives are sensitive to calendar definition, for instance bioclimatic indicators might be less dependent on the artificial definition of seasons, a typical example here is the the growing degree-days (GDD). In addition, we examined the influence of the calendar effect on the simulated vegetation. For this we analyzed the simulated leaf area index. As revealed by Fig. S13, even during boreal autumn, the deviation in leaf area index between classical and angular calendars is below 0.06% for PI and MH, and below 0.2% for LIG. Therefore, the calendar effect plays no significant role for this vegetation-related variable.*

6. In section 3.4, why only analyze the temperature over continents and ice-free continents? I would suggest to add analysis also on temperature over ocean and ice continents, which would be particularly important for Southern Hemisphere.

**Thanks for the constructive comment, we re-plotted the figures to include also the ice sheet regions. In addition, we add two more figures to show the calendar effect on Northern and Southern Hemisphere oceans. We refer to section 3.4 of our revised paper.**

Specific comments:

Line 25: change "31" to "21"; is there any other reference date that has been used by other model groups? Is there a reason to use March 21 or other date as the reference date?

**Thanks for the correction. We have corrected the date for vernal equinox in the revised manuscript. The reason to use 21th March as vernal equinox is because it is the day that the Sun is exactly above the Equator therefore our Earth have equal length of day and night. According to the reviewer's comment, we add in the revised manuscript (L27-29 of the revised manuscript and also L29-30 of the difference-tracked version):**

*In most models the vernal equinox is set to 21th March, when the Sun is exactly above the Equator which leads to equal length of day and night on our Earth.*

L32: change "with a periodicity of about 100,000 years" to "with major periodicities of about 400,000 and 100,000 years (Berger, 1978, https://doi.org/10.1175/1520-0469(1978)035<2362:LTVODI>2.0.CO;2; Berger and Loutre, 1991, https://doi.org/10.1016/02 3791(91)90033-Q)". The 400 ka periodicity is more important than the 100 ka one.

**Thanks for the correction and references, we have made the changes accordingly (L36-37 of the revised manuscript, L36-37 of the difference-tracked version).**

L33: what does "periods of perihelion and aphelion" mean?

**Sorry for the confusing, we now have made it clearer in the revised version (L35 of the revised manuscript, L37-39 of the difference-tracked version):**

*When the eccentricity is large, there is also a big difference between the perihelion distance and the aphelion distance, while at a small eccentricity when the orbit is more circular this difference is less pronounced.*

L34: 0.0167, 0.0189, and 0.0397 are the values at a given date or an average over a given period? Please make it clear in the manuscript.

**We made it clearer in our revised manuscript (L37-38 of the revised manuscript, L39-41 of the difference-tracked version):**

*Earth's orbital eccentricity is 0.016764, 0.018682, and 0.039378 in 1850 CE (pre-industrial), 6 ka B.P. (mid-Holocene) and 127 ka B.P. (Last interglacial) respectively.*

L37: with a "major" period of 41,000 years

**Thanks for the correction, we added the term "major" in our revised manuscript. (L41 of the revised manuscript, L44 of the difference-tracked version)**

L39: Note that 19,000 and 23,000 years are the major periodicities of climatic precession, not of astronomical precession. Please clarify it in the manuscript and cite Berger (1978) where these periodicities were calculated for the first time.

**Thanks for the correction, in the revised paper we changed the texts into (L42-44 of the revised manuscript, L44-46 of the difference-tracked version):**

*At the same time, the wobble of Earth's rotational axis (precession) modifies the direction of the Earth's tilt and determines which hemisphere is tilted towards the sun at perihelion. The major periodicities of climatic precession are around 19,000 and 23,000 years (Berger, 1978).*

L46: Indicating 6ka for MH and 127 ka for LIG could be misleading, because MH and LIG are more than these two dates.

**Based on the reviewer's comment we have corrected in our manuscript (L51-52 of the revised manuscript, L54-55 of the difference-tracked version):**

*Two interglacial episodes, i.e., the mid-Holocene (MH, a period roughly from 7 to 5 ka B.P) and the Last Interglacial (LIG, roughly equivalent to 130-115 ka B.P.), are particularly the focus of PMIP.*

155    L51: Herold et al 2012 (https://doi.org/10.1016/j.quascirev.2012.08.020 ) and Nikolova et al 2013 (doi:10.5194/cp-9-1789-2013 ) are among the early studies on simulating the 127ka climate and describing its insolation pattern, and deserve to be cited here. Please also be more precise about "enhanced seasonal cycles". In Nikolova et al 2013, it is found a larger seasonal contrast in northern hemisphere but a reduced one in the southern hemisphere.

**Thanks. We have added the two references in the updated version. Comparing Fig. 2 and Fig. 4 in Nikolova et al.**
160    **(2013) for the JJA and DJF surface temperature anomalies between MIS5e and PI, we found both CCSM3 and LOVE-CLIM present a larger seasonal contrast during MIS5e than PI over most areas of the Southern Hemisphere, especially over the continents, though such effect is less pronounced over Southern Hemisphere oceans in CCSM3. In addition we also find smaller seasonality across specific regions like the Lazarev Sea (in CCSM3) and South Atlantic Ocean (in LOVECLIM).**

165    **According to the reviewer's comment, we updated the texts (L54-60 of the revised manuscript, L58-65 of the difference-tracked version):**

*Due to the Earth's orbital parameter anomalies with respect to the present, the MH and LIG receive more insolation in summer and less in winter over the Northern Hemisphere, leading to larger seasonal contrast in the two time periods, which holds true for both hemispheres in most model simulations (Kukla et al., 2002; Shi and Lohmann, 2016; Shi et al., 2020;*
170    *Zhang et al., 2021; Kageyama et al., 2021; Herold et al., 2012). Such effect is much more profound in the LIG than in the MH (Lunt et al., 2013; Pfeiffer and Lohmann, 2016). However, in earlier simulations using CCSM3 and LOVECLIM, Nikolova et al. (2013) found smaller seasonality across Lazarev Sea (in CCSM3) and South Atlantic Ocean (in LOVECLIM) during the Last interglacial as compared to PI.*

L67: "we perform . . ." is unclear for me. Please give more explanation.

175    **In the revised version, we described (L79-82 of the revised manuscript, L84-88 of the difference-tracked version):**

*In Bartlein and Shafer (2019), the "pure" calendar effects have been examined by applying the angular calendar of 6 ka, 97 ka, 116 ka, and 127 ka onto modern observations. In the present study, we perform a calendar adjustment based on the actual past time intervals of the different model experiments. In detail, we apply an angular calendar of 0 ka, 6 ka, and 127 ka for the pre-industrial, mid-Holocene, and Last interglacial simulation respectively.*

180    Section 2.1: Is the calendar conversion method used in this study similar to those used in the five studies mentioned in my major comment 1?

**The methods adopted in our study is different, we have written in the discussion (please also see our response to major comment 1):**

*In previous studies, the angular calendar was defined using the true anomaly of the Earth corresponding to the present-*
185    *day seasons, in other words, each month begins and ends at the same celestial longitude as present-day for any period (Joussaume and Braconnot, 1997; Bartlein and Shafer, 2019; Timm et al., 2008; Chen et al., 2011; Pollard and Reusch, 2002). The work of Chen et al. (2011) and Timm et al. (2008) applied a 360-day year which is, originally, divided into 12 months with 30 days. The VE is set to day 81 in a calendar year. Pollard and Reusch (2002), Joussaume and Braconnot*

*(1997) and Bartlein and Shafer (2019), on the other hand, performed the calendar adjustment based on today's classical*

190 *calendar with 365 days in a non-leap year. In their studies, an assumption was made that the seasonality defined by the classical calendar is in phase with the insolation and solar geometry for modern-day. In our study, by calculating the onset of present-day months/seasons using the approach described in Section 2.1, we find that the classical "fixed-length" calendar is very similar to the angular calendar for today, but they are not completely the same. This is evidenced in the small shift of months between the two calendars as seen in Table 3. In particular the angular October is delayed by 3 days*

195 *compared to the classical October, resulting in negative anomalies in the adjusted-minus-unadjusted solar insolation.*

L79: is it necessary to mention "Northern Hemisphere"?

**The equinox of the Southern Hemisphere is called "the autumnal or fall equinox" and the "vernal equinox" is only for the Northern Hemisphere. So we agree with the reviewer that the term "Northern Hemisphere" is not necessary here. We have deleted it from the revised manuscript.**

200 L85: "orbital period" is confusing. I assume T is the number of days in one year.

**Orbital period is the Earth's revolution period, i.e., the time it takes the earth to revolve around the sun. So T is equal to 1 year or 365 days. To make it clearer in the paper, we have updated the texts (L101-102 of the revised manuscript, L107-108 of the difference-tracked version):**

*Here, $t_p$ denotes the time elapsed since Earth passes the perihelion and $T$ is the Earth's revolution period (i.e., 1 year or*

205 *365 days), namely the time it takes the Earth to make one complete revolution around the sun.*

Equation 2: Please explain what is the principle on which Equation (2) is built.

**Equation 2 refers to: $E - \epsilon \cdot \sin(E) = M$, it is called Kepler's equation and is based on Keplers' 1st and 2nd laws. Now we clarify this point in the manuscript (L104-106 of the revised manuscript, L110-102 of the difference-tracked version):**

210 *Equation (2) is called Kepler's equation and is based on Kepler's 1st and 2nd laws. The first law simply states that the orbit of a planet is an ellipse with the Sun at one of the two focus points, and the Kepler's 2nd law states that a line segment connecting the sun and a planet sweeps out equal areas during equal intervals of time.*

Equation 3: It is unclear how equation (3) is obtained. Please also give an equation explicitly relating tp to true anomaly.

**Equation 3 is obtained from Eq. 3.13b of Curtis (2014) which is based on trigonometric calculations. In detail:**

215 **In Fig. 3, starting with the observation that cos(E)=|ZS|/|ZQ|, which follows directly from the definition of cos. That means: |ZS|=cos(E)*|ZQ|=cos(E)*a, where a is the length of the semi-major axis. That gives us a representation of |ZS| in dependence of the eccentric anomaly E. As we want to relate E to the true anomaly $\theta$, we now want to express |ZS| in terms of $\theta$.**

**We start by simply observing that:**

$$|ZS| = |ZF| - |FS| = \epsilon a - |FS| \tag{1}$$

[Figure]

**Figure 3.** The geometric relationship between the eccentric anomaly E and the true anomaly $\theta$.

**|ZF|=$\epsilon$a by the definition of the eccentricity $\epsilon$.**

**The angle $\angle$ZFQ at F is given by $\pi$ - $\theta$. Applying the cos at this angle and taking into account that cos($\pi$ - $\theta$) = - cos($\theta$), we gain:**

$$|FS| = -cos(\theta) * |FP| = -cos(\theta) * r \tag{2}$$

**|FP| = r, the distance between the Earth and the sun. Plugging Eq. 2 in Eq. 1 yields:**

$$|ZS| = \epsilon a + r * cos(\theta) \tag{3}$$

**Therefore we know that:**

$$a * cos(E) = \epsilon a + r * cos(\theta) \tag{4}$$

**Which gives the desired relation between the eccentric anomaly E and the true anomaly $\theta$. This equation looks totally different to equation (3) in the manuscript - however, from here on it is only math and handling trigonometric equations to get the equation.**

**Then we have an explicit representation of tp as a function of $\theta$. It looks like:**

$$t_p(\theta) = \frac{MT}{2\pi} = \frac{(E - \epsilon \cdot \sin(E))T}{2\pi} \text{ with } E = 2 \cdot \arctan\left(\sqrt{\frac{1-\epsilon}{1+\epsilon}} \cdot \tan(\tfrac{\theta}{2})\right)$$

**In our revised paper, we add reference of Curtis (2014), and add one equation of how tp and $\theta$ are related (we refer to Eq. 4 in our manuscript).**

L94-96: Please be more precise on how each season is defined from the true anomaly that is calculated in equation (3).

**We agree with the reviewer's comment that it can make things clearer if we describe it more detailed using the formula. In our updated manuscript the following texts are added (L111-129 of the revised manuscript and L117-135 of the difference-tracked version):**

*The relation between the true anomaly $\theta$ and the time elapsed since Earth passes perihelion $t_p$ allows to define seasons with respect to Earth's position on the orbit rather than relying on a fixed number of days. Based on the "fixed-angular" approach, there are two ways to define the seasons: 1) The orbit is distinguished into four segments: A true anomaly of $\theta = 0°$) corresponds to March 21st and therefore marks the first day of spring. The length of the summer is gained by calculating $t_p$ ($\theta = 90°$). Similarly, the terms $t_p$ ($\theta = 180°$) and $t_p$ ($\theta = 270°$) mark the beginning of fall and winter, respectively. 2) The other method is based on the "meteorological" definition, in which the spring is defined as March-April-May, as typically done in paleoclimate modelling, although the VE is set to March 21st. The second approach is adopted in our study, and in this case, we firstly compute the starting and end time for each month, then average over the respective months in order to compare the angular seasonal means with the classical seasonal means. Months can be defined as $30°$ increments of the true anomaly. Just one additional step has to be executed before calculating angular months: As no months starts at the VE, the starting day has to be shifted from March 21st to April 1st. Since the time between today's March 21st and April 1st may not be true for past calendars, we defined April 1st by the angle. Therefore, we first calculate the angle between today's March 21st, noon (the VE) and the point of time occurring 10.5 days later, denoting April 1st. Finally, starting from the angle corresponding to April 1st, we are able to calculate the starting time of the next month by $30°$ increments of the true anomaly. Here we apply the so-called "largest remainder method": the number of days defined by the $30°$ of true longitude usually consists of an integer part plus a fractional remainder. Each month is firstly allocated a number of days equal to its respective integer part (for example, if January has 31.76 days, 31 days are allocated). This generally leave some days unallocated. The months are then ranked according to their fractional remainders, then an additional day is allocated to each of the months with the largest remainders until all days have been allocated.*

L119: you mean "increased summer insolation"?

**The Northern Hemisphere insolation during 6 ka is larger as compared to today for both summer and annual mean. Actually here we mean the annual mean insolation, as we are comparing the annual mean surface air temperature between 6 ka and PI. We now make it clearer in the updated paper (L169 of the revised manuscript and L178 of the difference-tracked version).**

L124: Please explain what is the delayed effect.

**We added in the revised version (L175-178 of the revised manuscript and L184-187 of the difference-tracked version):**
**Due to the large heat capacity of water, the ocean responses much more slowly to changes in incoming insolation than the land. Therefore, changes in solar radiation and surface air temperature over the oceans are out of phase. During**

265 **the MH, the Southern Hemisphere receives more radiation flux in SON in relative to present-day, leading to a warming of the Southern Ocean in DJF.**

L143: It is unclear how this calculation has been done, and how the dates in table 2 are obtained. More explanation is needed.

**I think this is a follow-up comment to a previous comment related to L94-96 (i.e., Please be more precise on how each season is defined from the true anomaly that is calculated in equation 3.) Here we give a more detailed explanation (can**
270 **also be seen in section 2.1, L119-129 of the revised manuscript and L125-135 of the difference-tracked version):**

*Months can be defined as 30° increments of the true anomaly. Just one additional step has to be executed before calculating angular months: As no months starts at the VE, the starting day has to be shifted from March 21st to April 1st. Since the time between nowadays March 21st and April 1st may not be true for past calendars, we defined April 1st by the angle. Therefore, we first calculate the angle between nowadays March 21st, noon (the VE) and the point of time occurring*
275 *10.5 days later, denoting April 1st. Finally, starting from the angle corresponding to April 1st, we are able to calculate the starting time of the next month by 30° increments of the true anomaly. Here we apply the so-called "largest remainder method": the number of days defined by the 30° of true longitude usually consists of an integer part plus a fractional remainder. Each month is firstly allocated a number of days equal to its respective integer part (for example, if January has 31.76 days, 31 days are allocated). This generally leave some days unallocated. The months are then ranked according to*
280 *their fractional remainders, then an additional day is allocated to each of the months with the largest remainders until all days have been allocated.*

L145-146: isn't the velocity always greater at perihelion than at aphelion, whatever at today or any time in the past?

**Yes we agree with the reviewer, and we updated our texts (L199-202 of the revised manuscript and L208-211 of the difference-tracked version):**
285 *Since the orbital velocity of the Earth is greater at perihelion than at aphelion, the seasons at aphelion are longer than at perihelion, for example for the present-day we have fewer days in winter and more days in summer, which is reflected both in today's classical calendar (DJF: 90 days; JJA: 92 days) and in the angular calendar (DJF: 89 days; JJA: 93 days).*

L235: It is good to refer to Rymes and Myers 2001 when use monthly values for calendar corrections, but it would be helpful for the reader if some information is given on how to transform monthly values to daily values.

290 **Thanks for the comment, now we have added the following (L304-311 of the revised manuscript and L314-321 of the difference-tracked version):**

*Daily output takes up much more space than monthly output, so most modelling groups only provide monthly frequency variables. Here, we utilize a calendar transformation method that requires only the raw (i.e., classical "fixed-length" calendar) monthly mean values (Rymes and Myers, 2001). In the study of Rymes and Myers (2001) an approach has been*
295 *introduced for smoothly interpolating coarsely-resolved data onto a finer resolution, while preserving the deterministic mean. Based on the approach, daily data can be reconstructed using the monthly mean values: The daily data is initialised with the monthly average of the respective month. Then, for each day of the year, its value is recursively recalculated as the*

*average of its own value and the values of the two adjacent days. After 365 iterations, this results in a nicely smooth annual cycle with the original monthly means being preserved.*

L280: any idea of why there is a model-dependency of the calendar effect?

**It is because the PI temperature response to calendar is not uniform, in the revised paper we clarified this point (L359-364 of the revised manuscript and L370-376 of the difference-tracked version):**

*In JJA, besides the slight cooling bias in the original mean surface air temperature for 6-3 ka as revealed by all the 3 models, we observe a model-dependency of the calendar effects for the time interval of 3-0 ka, during which the Northern Hemisphere classical mean temperature in JJA is underestimated by AWI-ESM and MPI-ESM, but very slightly overestimated by IPSL-CM. Such discrepancy between models is related to the spatially varying temperature changes over the Northern Hemisphere continents caused by the calendar effect (Fig. 1k).*

Section 4: discussion and conclusion are mixed up, better to be separated.

**Thanks for your comment. In the revised version we have separated discussion and conclusion sections. Here we refer to section 4 and 5 in our revised manuscript.**

L297: what does "the phasing of the insolation curve" mean? Does calendar effect have an influence on the phasing between insolation and temperature, precipitation in the Holocene transient simulation?

**Sorry for the confusion, we mean the angle of the orbit of the Earth around the Sun. For a classical "fixed-day" calendar, the insolation and temperature might be, to a certain degree, out of phase. This can be solved by using the angular calendar, defined on the angle of the orbit and the reference date. According to the comment, we now clarify in the paper (L381-383 of the revised manuscript and L396-398 of the difference-tracked version):**

*Two important elements should be taken into consideration when comparing paleoclimate simulations of different time intervals: the reference date (usually the VE), and the angle of the orbit of the Earth around the Sun, which defines the phasing of the insolation curve.*

L330-331: Please explain why a different definition of season is used in this study, is there any advantage?

**The most important reason is that today's "fixed-length" calendar, strictly speaking, is not an angular calendar, though it is very similar to the angular calendar for modern-day. If we perform a calendar correction based on the angles defined by present-day "fixed-length" calendar, as done in previous studies, a slight bias could be introduced for the angular calendar for past time periods, especially for boreal autumn.**

**In the revised paper, we illustrated that the angular calendar for PI has a shift of -1 to 3 days as compared to the classical calendar, therefore we also need to adjust the calendar for PI. The revised text is as follows (L412-416 of the revised manuscript and L440-444 of the difference-tracked version):**

*In our study, by calculating the onset of present-day months/seasons using the approach described in Section 2.1, we find that the classical "fixed-length" calendar is very similar to the angular calendar for today, but they are not completely the same. This is evidenced in the small shift of months between the two calendars as seen in Table 3. In particular the*

*angular October is delayed by 3 days compared to the classical October, resulting in negative anomalies in the adjusted-minus-unadjusted solar insolation.*

L352: The analysis of this study shows that the calendar effect is most important in autumn. Then would the model-proxy comparison be significantly affected if proxy records mainly reflect summer temperature?

335 **Most proxy records the summer or winter signal, in the discussion part we add two more paragraph to discuss about the implication of calendar on model-proxy comparison. The texts are as following (please see L436-458 of the revised manuscript and also L484-507 of the difference-tracked version):**

[revised manuscript text omitted]

This paper documents in detail the calendar effects on analyses of paleoclimate simulations for the mid-Holocene (MH), the Last Interglacial (LIG) and pre-industrial periods as well as for transient simulations. Indeed, due to the slow variations of the Earth's orbital parameters, the position of seasons is modified within the ellipse, affecting the length of the seasons. This effect has been documented in Joussaume and Braconnot (1997) and more recently in Bartlein and Shafer (2019) but is usually not accounted for. This paper uses the most recent simulations of PMIP4 with coupled models and allows to revisit this question. Indeed, at the time of PMIP1, in Joussaume and Braconnot (1997), sea surface temperatures were prescribed to today, thus including a hidden present-day calendar in past climate simulations. Moreover, the paper presents results from transient simulations. These results deserve to be published although some improvements of the text would help its readability.

**Dear Reviewer,**

**Thank you very much for your positive and constructive comments. In the following, we present our point-to-point responses. Our answers to your comments are written in bold.**

**Thanks again for your time and efforts.**

**Best,**

**Xiaoxu**

There is a need to better explain how the 90° angular seasons are positioned relative to the vernal equinox which provides the reference for dates (March 21st), the way the seasons are computed is not fully clear in the paper.

To answer this question, we first add an equation which relates the time elapsed since Earth passes the perihelion to true anomaly $\theta$ (the angle between the axis of the perihelion and the actual position of the earth):

$$t_p(\theta) = \frac{MT}{2\pi} = \frac{(E - \epsilon \cdot \sin(E))T}{2\pi} \text{ with } E = 2 \cdot \arctan\left(\sqrt{\frac{1-\epsilon}{1+\epsilon}} \cdot \tan\left(\frac{\theta}{2}\right)\right)$$

Then we clarify in our revised paper (please see L111-119 of the revised manuscript and also L117-125 of the difference-tracked version):

*The relation between the true anomaly $\theta$ and the time elapsed since Earth passes perihelion $t_p$ allows to define seasons with respect to Earth's position on the orbit rather than relying on a fixed number of days. Based on the "fixed-angular" approach, there are two ways to define the seasons: 1) The orbit is distinguished into four segments: A true anomaly of $\theta = 0°$) corresponds to March 21st and therefore marks the first day of spring. The length of the summer is gained by calculating $t_p$ ($\theta = 90°$). Similarly, the terms $t_p$ ($\theta = 180°$) and $t_p$ ($\theta = 270°$) mark the beginning of fall and winter, respectively. 2) The other method is based on the "meteorological" definition, in which the spring is defined as March-April-May, as typically done in paleoclimate modelling, although the VE is set to March 21st. The second approach is adopted in our study, and in this case, we firstly compute the starting and end time for each month, then average over the respective months in order to compare the angular seasonal means with the classical seasonal means.*

We also add texts on how we define months (please see L119-129 of the revised manuscript and also L126-135 of the difference-tracked version):

*Months can be defined as $30°$ increments of the true anomaly. Just one additional step has to be executed before calculating angular months: As no months starts at the VE, the starting day has to be shifted from March 21st to April 1st. Since the time between nowadays March 21st and April 1st may not be true for past calendars, we defined April 1st by the angle. Therefore, we first calculate the angle between nowadays March 21st, noon (the VE) and the point of time occurring 10.5 days later, denoting April 1st. Finally, starting from the angle corresponding to April 1st, we are able to calculate the starting time of the next month by $30°$ increments of the true anomaly. Here we apply the so-called "largest remainder method": the number of days defined by the $30°$ of true longitude usually consists of an integer part plus a fractional remainder. Each month is firstly allocated a number of days equal to its respective integer part (for example, if January has 31.76 days, 31 days are allocated). This generally leave some days unallocated. The months are then ranked according to their fractional remainders, then an additional day is allocated to each of the months with the largest remainders until all days have been allocated.*

Concerning the simulations used, it would be good to summarize the PMIP4 boundary conditions in section 2.2 and explicit whether the three transient simulations use the same boundary conditions and whether they differ or not with PMIP4 for the mid-Holocene.

According to the comment, we add a table describing the PMIP4 boundary conditions for pre-industrial, mid-Holocene and Last interglacial (see table 1 of this letter), and we also illustrated in the paper (please see table 2 in the paper, L138-145 of the revised manuscript and also L147-154 of the difference-tracked version):

**Table 1.** PMIP4 boundary conditions for pre-industrial, mid-Holocene and Last interglacial.

| Experiment | CO$_2$ (ppm) | CH$_4$ (ppb) | N$_2$O (ppb) | Eccentricity | Obliquity | perihelion - 180° |
|---|---|---|---|---|---|---|
| PI | 284.3 | 808.2 | 273 | 0.016764 | 23.459° | 100.33° |
| MH | 264.4 | 597 | 262 | 0.018682 | 24.105° | 0.87° |
| LIG | 275 | 685 | 255 | 0.039378 | 24.040° | 275.41° |

*According to Otto-Bliesner et al. (2017), the CO$_2$ concentration applied in the PMIP4 protocol for mid-Holocene is derived from ice-core measurements from Dome C (Monnin et al., 2001, 2004). CH$_4$ has been derived from multiple Antarctic ice cores including EPICA Dome C (Flückiger et al., 2002), EPICA Dronning Maud Land (Barbante et al., 2006) and Talos Dome (Buiron et al., 2011).The N$_2$O data around 6 ka are compiled from EPICA Dome C (Flückiger et al., 2002; Spahni et al., 2005) and Greenland ice cores. The concentrations of CO$_2$ during the LIG are derived from Antarctic ice cores (Bereiter et al., 2015; Schneider et al., 2013), CH$_4$ has been derived from EPICA Dome C and EPICA Dronning Maud Land (Loulergue et al., 2008; Schilt et al., 2010b), and N$_2$O from EPICA Dome C and Talos Dome (Schilt et al., 2010b, a). Table 2 provides a summary of PMIP4 boundary conditions for pre-industrial, mid-Holocene and Last interglacial.*

**Then we add another sentence in our paper (please see L160-163 of the revised manuscript and also L170-172 of the difference-tracked version):**

*Therefore, in the transient simulations, the orbital forcings used at 6 ka and 0 ka are the same as the PMIP4 equilibrium simulations. However, there are differences between the greenhouse gas concentrations applied in the transient and PMIP4 equilibrium simulations, as the values have been taken from different reconstructions.*

The use of angular seasons is indeed more appropriate when comparing seasons from different periods in paleoclimate simulations, however, it would be good to add some elements in the discussion on possible implications for the model-proxy data comparisons.

**It is a good idea to discuss about the implication of calendar on model-proxy comparison, based on the comment, we added in the discussion section (please see L436-458 of the revised manuscript and also L484-507 of the difference-tracked version):**

*Proxy-based reconstructions provide us another ability to examine the temperature evolution of the past and can help assess the model's performance in simulating the past climates. Since paleoclimate data often records the seasonal signal (e.g. local summer temperature), an appropriate choice of calendar is therefore important for temperature comparisons between model results and proxy data. For the mid-Holocene, Bartlein et al. (2011) is an often-cited study that compiled pollen-based continental temperature reconstructions. The question arises whether the consideration of calendar effects could lead to an improved model-data agreement. Here we show in Fig. S11 the simulated classical mean temperature anomalies (MH minus PI) versus continental reconstructions. The expected increased seasonality occurs only over Northwest Europe as indicated by the proxy records. The opposite sign is shown over northern America, with winter warming and summer cooling, and is therefore not consistent with the ensemble model result. Bartlein et al. (2011) attributes such a*

*model-data mismatch to changes in local atmospheric circulation that tend to overwhelm the insolation effect. The calendar impacts, as illustrated in Fig. 4, result in warming of less than 0.2 K over the Northern Hemisphere in both DJF and JJA, implying that model-data consistency is improved for Northwest Europe in summer, and Northern America in winter, while for most other regions using the adjusted calendar results in a poorer match between model and proxy temperatures. These results reveal that for the mid-Holocene the calendar adjustment does not guarantee a better model-data agreement, and the underlying reason might be that, in addition to the solar insolation, the proxy could be strongly influenced by the local environment, such as flow of humid air and increased cloud cover (Harrison et al., 2003) or warm-air advection (Bonfils et al., 2004).*

*Since there are very few high-resolution reconstructed temperature records for the LIG, we use here the compilation from Turney and Jones (2010) for the annual mean temperature anomalies between LIG and PI, and compare them with modeled classical mean values for boreal summer (Fig. S12). We keep in mind that the summer mean LIG temperatures are usually higher than the annual mean values documented by the proxy records. At high latitudes of Northern Hemisphere continents (e.g. Greenland, Russia and Alaska), as well as over subpolar oceans (e.g. the Nordic Sea and the Labrador Sea), we find that the models underestimate the recorded LIG warming. Part of the bias can be corrected by calendar adjustment which leads to a warming of up to 1 K over Northern Hemisphere continents in JJA (Fig. 3k).*

Moreover, you have made the choice to define 4 times 90° angular seasons and to compare to pre-industrial, but nothing is said on how you would compare to present-day. In Joussaume and Braconnot (1997), the choice was to use the same angular seasons as used today (even if they are not perfect 90° angles) to ensure consistency with present-day. It could be interesting to add some discussion on the impact of those different choices.

**Thanks for the suggestion. Now in our revised manuscript we added the following texts into the discussion part (we also refer to L405-428 of the revised manuscript and L434-462 of the difference-tracked version):**

*In previous studies, the angular calendar was defined using the true anomaly of the Earth corresponding to the present-day seasons, in other words, each month begins and ends at the same celestial longitude as present-day for any period (Joussaume and Braconnot, 1997; Bartlein and Shafer, 2019; Timm et al., 2008; Chen et al., 2011; Pollard and Reusch, 2002). The work of Chen et al. (2011) and Timm et al. (2008) applied a 360-day year which is, originally, divided into 12 months with 30 days. The VE is set to day 81 in a calendar year. Pollard and Reusch (2002), Joussaume and Braconnot (1997) and Bartlein and Shafer (2019), on the other hand, performed the calendar adjustment based on today's classical calendar with 365 days in a non-leap year. In their studies, an assumption was made that the seasonality defined by the classical calendar is in phase with the insolation and solar geometry for modern-day. In our study, by calculating the onset of present-day months/seasons using the approach described in Section 2.1, we find that the classical "fixed-length" calendar is very similar to the angular calendar for today, but they are not completely the same. This is evidenced in the small shift of months between the two calendars as seen in Table 3. In particular the angular October is delayed by 3 days compared to the classical October, resulting in negative anomalies in the adjusted-minus-unadjusted solar insolation. Though different methods are used in our work from the mentioned previous studies, our results are identical: for the LIG, the*

*adjusted-minus-unadjusted surface air temperature over the Northern Hemisphere is up to 5 K during SON (Joussaume and Braconnot, 1997; Bartlein and Shafer, 2019; Chen et al., 2011) or September (Pollard and Reusch, 2002); and the Northern Hemisphere monsoon precipitation in SON is underestimated by the use of the classical calendar (Bartlein and*
115 *Shafer, 2019; Chen et al., 2011). Similar biases are found for the early-Holocene (Timm et al., 2008) and mid-Holocene (Joussaume and Braconnot, 1997; Bartlein and Shafer, 2019) but less pronounced . These results are consistent with the findings in our study, however, comparing results of our 3 transient simulations with that from the TraCE-21ka transient simulation, as it was investigated in Bartlein and Shafer (2019), distinct differences emerge for the boreal autumn surface air temperature near present-day. In Bartlein and Shafer (2019), the artificial bias in MH-minus-PI temperature and pre-*
120 *cipitation totally stems from the bias in MH when the classical calendar is applied (as for PI both calendars are identical). In contrast, our study reveals that such bias is mainly dominated by the deviation between angular and classical calendars for present-day. It should be noted that these discrepancies are not due to the different models used in our studies, but rather to the different approaches adopted for calendar adjustment.*

In the discussion, it would also be interesting to know if some of the forcing or boundary conditions of simulations may still
125 keep some memory of the present-day calendar (e.g., prescribed vegetation or aerosols) and may add some bias in the analyses.

**Thanks for the interesting comment, we now discussed about this point in the revised version (please also refer to L465-469 of the revised manuscript and L535-537 of the difference-tracked version):**

*Finally, we should bear in mind that the forcing or boundary conditions of simulations may still keep some memory of the present-day calendar (e.g., prescribed ozone, vegetation or aerosols). This is particularly important for paleoclimate*
130 *simulations with stand-alone atmosphere or ocean models, as they are often forced by fields in classical calendar, and this may introduce further bias in the simulated seasonality even though with the calendar being adjusted.*

The text needs some reading to correct some sentences, some are mentioned below.

Specific comments:

L12: The largest difference occurs in autumn is related to the choice of a fixed date for the vernal equinox, this should be
135 made clearer in this sentence

**Thanks for the suggestion, we now add in our manuscript (L11-13 of the revised manuscript, L12-13 of the difference-tracked version):**

*The largest cooling bias occurs in boreal autumn when the classical calendar is applied for the mid-Holocene and last interglacial, due to the fact that the vernal equinox is fixed at 21th March.*

140 L16: the conclusion on using monthly data is not clear in the abstract, you should add compared to using daily data

**To make it clearer, now we illustrated in the revised version (L16-18 of the revised manuscript, L16-19 of the difference-tracked version):**

*Finally, monthly-adjusted values for surface air temperature and precipitation are very similar to the daily-adjusted values, therefore correcting the calendar based on the monthly model results can largely reduce the artificial bias.*

145 L24: "is highly depends" should be "highly depends"

**Thanks for the correction. We have corrected the error in the revised version.**

L25: March 21st and not 31 !

**Thanks for the correction. We have corrected the date for vernal equinox in the revised manuscript.**

L30: the classical reference is rather to Berger (1978) than 1977

150 **Thanks for the correction, we now refer to Berger (1978) in our revised version.**

L44: modelling groups and not model groups

**We changed it into *"a number of modelling groups"***

L48: I do not think we can say that the MH and LIG are chosen due to their great potential to resemble future scenarios. Please reconsider this statement

155 **We now changed the texts in the revised paper (L51-53 of the revised manuscript, L54-56 of the difference-tracked version):**

*Two interglacial episodes, i.e., the mid-Holocene (MH, a period roughly from 7 to 5 ka B.P) and the Last Interglacial (LIG, roughly equivalent to 130-115 ka B.P.), are particularly the focus of PMIP (Otto-Bliesner et al., 2017), as they are the two most recent warm periods in geological history.*

160 L50: receive more insolation in summer and less in winter is only true for the Northern Hemisphere

**We agree. Based on the comment, we changed in our manuscript (L54-60 of the revised manuscript, L58-65 of the difference-tracked version):**

*Due to the Earth's orbital parameter anomalies with respect to the present, the MH and LIG receive more insolation in summer and less in winter over the Northern Hemisphere, leading to larger seasonal contrast in the two time periods, which*

165 *holds true for both hemispheres in most model simulations (Kukla et al., 2002; Shi and Lohmann, 2016; Shi et al., 2020; Zhang et al., 2021; Kageyama et al., 2021; Herold et al., 2012). Such effect is much more profound in the LIG than in the MH (Lunt et al., 2013; Pfeiffer and Lohmann, 2016). However, in earlier simulations using CCSM3 and LOVECLIM, Nikolova et al. (2013) found smaller seasonality across Lazarev Sea (in CCSM3) and South Atlantic Ocean (in LOVECLIM) during the Last interglacial as compared to PI.*

170 L83 to L86: a drawing to explain M and E is missing. It could be added at least in the supplementary material

**Based on the comment, we added a figure in the supplementary describing the relation among he mean anomaly (M), eccentric anomaly (E) and true anomaly ($\theta$), here we refer to Fig. S1 in the revised supplementary.**

L102: IPSL is the name of the institution not of the model (IPSL-CM)

**Thanks for the correction, we now changed the name of the model into "IPSL-CM" in the revised manuscript.**

175     L132: Sahel and not Sahal

**Sorry for the typo. We now corrected this term in the updated manuscript.**

Table 2: mentions that the present-day calendar is not an angular one and should be corrected: could that correction be described ? at least in supplementary ?

**In previous studies (e.g. Joussaume and Braconnot (1997)), the choice was to use the same angular seasons as used**
180     **today, though they are not perfect 90° angles. In our study, we would like to calculate the start of each season/month according to an accurate 90° /30° increment of the true anomaly. In the revised version, we described how we performed the calendar correction for present-day in more detail (L111-129 of the revised manuscript and L117-135 of the difference-tracked version):**

*The relation between the true anomaly $\theta$ and the time elapsed since Earth passes perihelion $t_p$ allows to define seasons*
185     *with respect to Earth's position on the orbit rather than relying on a fixed number of days. Based on the "fixed-angular" approach, there are two ways to define the seasons: 1) The orbit is distinguished into four segments: A true anomaly of $\theta = 0°$) corresponds to March 21st and therefore marks the first day of spring. The length of the summer is gained by calculating $t_p$ ($\theta = 90°$). Similarly, the terms $t_p$ ($\theta = 180°$) and $t_p$ ($\theta = 270°$) mark the beginning of fall and winter, respectively. 2) The other method is based on the "meteorological" definition, in which the spring is defined as March-*
190     *April-May, as typically done in paleoclimate modelling, although the VE is set to March 21st. The second approach is adopted in our study, and in this case, we firstly compute the starting and end time for each month, then average over the respective months in order to compare the angular seasonal means with the classical seasonal means. Months can be defined as 30° increments of the true anomaly. Just one additional step has to be executed before calculating angular months: As no months starts at the VE, the starting day has to be shifted from March 21st to April 1st. Since the time*
195     *between today's March 21st and April 1st may not be true for past calendars, we defined April 1st by the angle. Therefore, we first calculate the angle between today's March 21st, noon (the VE) and the point of time occurring 10.5 days later, denoting April 1st. Finally, starting from the angle corresponding to April 1st, we are able to calculate the starting time of the next month by 30° increments of the true anomaly. Here we apply the so-called "largest remainder method": the number of days defined by the 30° of true longitude usually consists of an integer part plus a fractional remainder. Each*
200     *month is firstly allocated a number of days equal to its respective integer part (for example, if January has 31.76 days, 31 days are allocated). This generally leave some days unallocated. The months are then ranked according to their fractional remainders, then an additional day is allocated to each of the months with the largest remainders until all days have been allocated.*

L155-156: you compare in the following angular (adjusted) versus calendar (non adjusted), as is chosen on the figures as
205     well, whereas in this sentence you reverse the comparison. Please take care to avoid changing the reference to help the reader.

**According to the reviewer's comment, we changed in our manuscript (L210-213 of the revised manuscript and L219-223 of the difference-tracked version):**

*Fig. 1 depicts the differences in seasonal surface air temperature between angular and classical means. Positive/negative values indicate warming/cooling in angular mean temperatures as compared to classical mean temperatures. We observe spatially-variable changes of surface air temperature in adjusted values as compared to unadjusted values.*

L180: It is expected to have continents reacting faster than oceans to solar forcing due to the differences in heat capacity.

**We totally agree and in the updated paper we wrote (L237-238 of the revised manuscript and L246 of the difference-tracked version):**

*the calendar effect on surface air temperature over the ocean is delayed due to the large heat capacity of sea water.*

L312: Use of daily data "can completely erase the bias" is strange, isn't it the definition of what is called the bias in the paper ? I guess you mean that compared to daily data, using monthly data do not completely erase the bias ?

**Sorry for the confusion, we mean that in order to remove the artificial bias, daily data is necessary for the calendar correction. Now in order to avoid confusion, we directly say in our manuscript (L390-391 of the revised manuscript and L412-414 of the difference-tracked version):**

*Daily data is needed for calendar adjustment, however, due to the large volume of daily outputs, they are not preserved by most modelling groups.*

L332: indeed, in Joussaume and Braconnot (1997) the choice is made to use the same seasons as defined today to be compatible with the present-day reference. It would be useful to discuss more the implication of your choice if you want to compare to today.

**Today's "fixed-length" calendar, strictly speaking, is not an angular calendar, though it is very similar to the angular calendar for modern-day. If we perform calendar correction based on present-day reference, as done in previous studies, slight bias could be introduced for the angular calendar for past time periods, especially for boreal autumn.**

**In the revised paper, we illustrated that the angular calendar for PI has a shift of -1 to 3 days as compared to the classical calendar, therefore we also need to adjust the calendar for PI. The texts are like the following (L412-416 of the revised manuscript and L440-444 of the difference-tracked version)**

*In our study, by calculating the onset of present-day months/seasons using the approach described in Section 2.1, we find that the classical "fixed-length" calendar is very similar to the angular calendar for today, but they are not completely the same. This is evidenced in the small shift of months between the two calendars as seen in Table 3. In particular the angular October is delayed by 3 days compared to the classical October, resulting in negative anomalies in the adjusted-minus-unadjusted solar insolation.*

L352: when considering proxy-data we may have to rather consider bioclimatic indicators which are less dependent on the artificial definition of seasons, eg when considering the growing degree-days

**We agree and discuss this point in the second last paragraph of the revised paper. In addition, we add another plot about the calendar effect on leaf area index, which shows that the leaf area index is not affected by the calendar**

240 **definition. In the discussion section, we add the following (L459-464 of the revised manuscript and also L508-513 of the difference-tracked version):**

*Not all types of archives are sensitive to calendar definition, for instance bioclimatic indicators might be less dependent on the artificial definition of seasons, a typical example here is the the growing degree-days (GDD). In addition, we examined the influence of the calendar effect on the simulated vegetation. For this we analyzed the simulated leaf area index. As*

245 *revealed by Fig. S13, even during boreal autumn, the deviation in leaf area index between classical and angular calendars is below 0.06% for PI and MH, and below 0.2% for LIG. Therefore, the calendar effect plays no significant role for this vegetation-related variable.*

Fig 3: legend of third column is angular minus classical and not classical minus angular

**Thanks for the correction, we updated the caption of Fig. 3 in the revised manuscript.**

**References**

[revised manuscript text omitted]

---

## Referee Report (RR1)

**Minor comments:**

Lines 26-29: I would suggest to remove the last point from the abstract ("One important…….and night on our Earth"). This is important information and should be provided in the manuscript, but it makes the abstract a bit too long. Furthermore, you mention this point already briefly on line 14.

Line 441: In figure S11 only the model-data comparison is shown for the classical calendar definition. Combining this with information from figure 4 the authors continue to argue that in some regions the model-data match is improved by using the angular-seasons, while in others it is deteriorated. This is all very descriptive and qualitative, why not actually show this in a figure?

Lines 452-458: Here LIG annual mean temperature anomaly reconstructions are compared with simulated summer temperature anomalies. I understand that the authors wish to present both model-data comparison for the MH and the LIG, but to me this LIG model-data comparison makes little sense. As the authors mention, annual mean temperature anomalies should not be impacted by the calendar definition. By comparing them with modelled summer temperature anomalies, a calendar effect can be shown, but is that sensible? Arguably, LIG annual mean temperature anomaly reconstructions include a seasonal bias, perhaps towards summer, but to me that seems outside of the scope of this manuscript. Perhaps simply leave out this section and concentrate on the MH model-data comparison?

Lines 459-464: It is not clear to me why 'bioclimatic indicators' would be less dependent on the calendar definition. With respect to the example shown here, I would expect that the leaf area index is strongly impacted by temperature, and since the authors show large SON changes in temperatures depending on the calendar definition, I would expect impacts on leaf area index as well. Shown here (fig S13) are percentages changes between the two calendar definitions for different periods and those are indeed small. But perhaps more importantly, how large are these differences compared to the differences between the different climatic periods? So for instance the SON leaf area index difference between PI and LIG? Are they of similar magnitude? In general, if an example like this is given, then sufficient information and discussion should be provided for the reader to follow and judge the line of reasoning.

Conclusion section: Given that this manuscript really covers all aspect and choices that need to be made when dealing with calendar issues in paleoclimate modelling work, the conclusion section seems to me to be a great place to clearly outline some "best practices". Which calendar-related adjustments should palaeo modellers do and which ones are of second-order importance?

Figure 11: There are some important differences between the seasonal cycles based on daily versus monthly precipitation data. This is not unexpected, but should be mentioned in the main text.

**Technical comments:**

Line 56: Perhaps "larger seasonal temperature contrast"? If it is not temperature but insolation that you are referring to, then I don't understand the remark about "which holds true for both hemispheres in most model simulations", because TOA insolation should indeed be nearly identical in models.

Line 105: "and Kepler's $2^{nd}$ law states"

Line 183: "more than"

Figure 11: some labels seem missing.

---

## Author Response (AR2)

**Response letter**

Xiaoxu Shi[1], Martin Werner[1], Carolin Krug[1,2], Chris M. Brierley[3], Anni Zhao[3], Endurance Igbinosa[1,2], Pascale Braconnot[4], Esther Brady[5], Jian Cao[6], Roberta D'Agostino[7], Johann Jungclaus[7], Xingxing Liu[8], Bette Otto-Bliesner[5], Dmitry Sidorenko[1], Robert Tomas[5], Evgeny M. Volodin[9], Hu Yang[1], Qiong Zhang[10], Weipeng Zheng[11], and Gerrit Lohmann[1,2]

[1]Alfred Wegener Institute, Helmholtz Center for Polar and Marine Research, Bremerhaven, Germany
[2]Bremen University, Bremen, Germany
[3]Department of Geography, University College London, London, UK
[4]Laboratoire des Sciences du Climat et de l'Environnement-IPSL, Unité Mixte CEA-CNRS-UVSQ, Université Paris-Saclay, Orme des Merisiers, Gif-sur-Yvette, France
[5]Climate and Global Dynamics Laboratory, National Center for Atmospheric Research (NCAR), Boulder, CO 80305, USA
[6]School of Atmospheric Sciences, Nanjing University of Information Science & Technology, Nanjing, 210044, China
[7]Max Planck Institute for Meteorology, Hamburg, Germany
[8]State Key Laboratory of Loess and Quaternary Geology, Institute of Earth Environment, Chinese Academy of Sciences, Xi'an, 710061, China
[9]Marchuk Institute of Numerical Mathematics, Russian Academy of Sciences, ul. Gubkina 8, Moscow, 119333, Russia
[10]Department of Physical Geography and Bolin Centre for Climate Research, Stockholm University, 10691, Stockholm, Sweden
[11]LASG, Institute of Atmospheric Physics, Chinese Academy of Sciences, Beijing, 100029, China

**Dear Reviewer,**

**Thank you very much for your positive and constructive comments. In the following, we present our point-to-point responses. Our answers to your comments are written in bold.**

**Thanks again for your time and efforts.**

5 **Best,**

**Xiaoxu**

**1 Comments from Reviewer 1**

Minor comment:

Lines 26-29: I would suggest to remove the last point from the abstract ("One important.......and night on our Earth"). This
10 is important information and should be provided in the manuscript, but it makes the abstract a bit too long. Furthermore, you mention this point already briefly on line 14.

**Thanks for the comment, we now have removed this part from the abstract.**

Line 441: In figure S11 only the model-data comparison is shown for the classical calendar definition. Combining this with information from figure 4 the authors continue to argue that in some regions the model-data match is improved by using the

15  angular-seasons, while in others it is deteriorated. This is all very descriptive and qualitative, why not actually show this in a figure?

**Thanks for the suggestion, according to the comment, we further updated Fig. S11, so that the plot includes model-data comparison for both calendars, in addition, we also add two panels on the calendar effects. Color-table has been adjusted to provide aid for readers with color blindness. For all other spatial plots in the present paper, we now use color-tables from https://www.ncl.ucar.edu/Document/Graphics/ColorTables/Aid$_{i}n_{c}olor_{b}lindness_{c}at.shtml$**

**Lines 452-458: Here LIG annual mean temperature anomaly reconstructions are compared with simulated summer temperature anomalies. I understand that the authors wish to present both model- data comparison for the MH and the LIG, but to me this LIG model-data comparison makes little sense. As the authors mention, annual mean temperature**

20  **anomalies should not be impacted by the calendar definition. By comparing them with modelled summer temperature anomalies, a calendar effect can be shown, but is that sensible? Arguably, LIG annual mean temperature anomaly reconstructions include a seasonal bias, perhaps towards summer, but to me that seems outside of the scope of this manuscript. Perhaps simply leave out this section and concentrate on the MH model-data comparison?**

**We agree that the LIG proxy might have a seasonal bias, according to the comment, we deleted the corresponding**

25  **texts from our updated manuscript.**

**Lines 459-464: It is not clear to me why 'bioclimatic indicators' would be less dependent on the calendar definition. With respect to the example shown here, I would expect that the leaf area index is strongly impacted by temperature, and since the authors show large SON changes in temperatures depending on the calendar definition, I would expect impacts on leaf area index as well. Shown here (fig S13) are percentages changes between the two calendar definitions for**

30  **different periods and those are indeed small. But perhaps more importantly, how large are these differences compared to the differences between the different climatic periods? So for instance the SON leaf area index difference between PI and LIG? Are they of similar magnitude? In general, if an example like this is given, then sufficient information and discussion should be provided for the reader to follow and judge the line of reasoning.**

**Thanks for the comment, we now re-plotted the figure in absolute values rather than percentage values, and we found**

35  **obvious calendar effect on Northern Hemisphere vegetation in LIG. We also add the following texts in the manuscript:**

*Since the calendar definition has a strong influence on the SON surface air temperatures, one might expect a clear response from the bioclimatic indicators, which are closely dependent on the environmental temperature. Here we investigate the influence of the calendar effect on the simulated vegetation. To do this, we analyzed the simulated leaf area index. As shown in Fig. S12, the leaf area index of the Northern Hemisphere during boreal autumn is evidently larger in angular*

40  *means than in classical means, suggesting that the definition of seasonality also has an impact on the vegetation pattern for LIG. However, for MH we do not observe significant changes in the leaf area index caused by calendar adjustment.*

**Conclusion section: Given that this manuscript really covers all aspect and choices that need to be made when dealing with calendar issues in paleoclimate modelling work, the conclusion section seems to me to be a great place to clearly**

outline some "best practices". Which calendar-related adjustments should palaeo modellers do and which ones are of second-order importance?

**We added in the conclusion:**

*Based on our results, we conclude that the necessity of calendar adjustment should depend on specific research content. It is crucial to perform such a seasonality correction when examining seasonal temperature and precipitation of the LIG. For MH, the calendar effect appears to be relatively minor during both DJF and JJA — two seasons that are frequently analyzed in paleoclimate studies. However, when it comes to the SON, the effect of the calendar on the surface air temperature of the MH is not negligible, so a calendar correction is necessary in this case.*

**Figure 11: There are some important differences between the seasonal cycles based on daily versus monthly precipitation data. This is not unexpected, but should be mentioned in the main text.**

**Thanks for the comment, now we have added in the texts:**

*In addition, we also observe some discrepancies between the seasonal cycles based on daily and monthly precipitation. One example is the peak value in July (late June) for MH (LIG) as indicated by the daily rainfall over South Asia, which is not presented in the monthly average. Similar cases can also be found for North America during warm months.*

**Technical comments:**

**Line 56: Perhaps "larger seasonal temperature contrast"? If it is not temperature but insolation that you are referring to, then I don't understand the remark about "which holds true for both hemispheres in most model simulations", because TOA insolation should indeed be nearly identical in models.**

**We agree, we now change the term into "larger seasonal temperature contrast"**

**Line 105: "and Kepler's $2^{nd}$ law states"**

**Thanks, we have modified the text accordingly.**

**Line 183: "more than"**

**Sorry for the typo, we have now corrected it.**

**Figure 11: some labels seem missing.**

**Thanks for the correction, we now have completed the legends in the plot.**

**2 Comments from Reviewer 2**

**Please change "perihelion-180" in Table 2 to "Longitude of perihelion".**

**Thanks for the correction, we have now changed the texts accordingly.**

**The description of Table 2 in the second paragraph of section 2.2 is not complete. In addition to the sources of Greenhouse gases concentrations, the reference for the orbital parameters should also be given (I assume it is Berger 1978).**

75      **Thanks for the comment, now we added in section 2.2:**

*The orbital parameters are calculated according to Berger (1978).*

[revised manuscript text omitted]